# TRANSFORMERS LEARN VARIABLE-ORDER MARKOV CHAINS IN-CONTEXT

## ABSTRACT

Large language models (LLMs) have demonstrated impressive in-context learning (ICL) capability. However, it is still unclear how the underlying transformers accomplish it, especially in more complex scenarios. Toward this goal, several recent works studied how transformers learn fixed-order Markov chains (FOMC) in context, yet natural languages are more suitably modeled by variable-order Markov chains (VOMC), i.e., context trees (CTs). In this work, we study the ICL of VOMC by viewing language modeling as a form of data compression and focusing on small alphabets and low-order VOMCs. This perspective allows us to leverage mature compression algorithms, such as context-tree weighting (CTW) and prediction by partial matching (PPM) algorithms as baselines, the former of which is Bayesian optimal for a class of priors that we refer to as the CTW priors. We empirically observe a few phenomena: 1) Transformers can indeed learn to compress VOMC in-context, while PPM suffers significantly; 2) The performance of transformers is not very sensitive to the number of layers, and even a two-layer transformer can learn in-context quite well; and 3) Transformers trained and tested on non-CTW priors can significantly outperform the CTW algorithm. To explain these phenomena, we analyze the attention map of the transformers and extract two mechanisms, on which we provide two transformer constructions: 1) A construction with $D + 2$ layers that can mimic the CTW algorithm accurately for CTs of maximum order $D$, 2) A 2-layer transformer that utilizes the feed-forward network for probability blending. These constructions can explain most of the phenomena mentioned above. One distinction from the FOMC setting is that a counting mechanism appears to play an important role. We implement these synthetic transformer layers and show that such hybrid transformers can match the ICL performance of transformers, and more interestingly, some of them can perform even better despite the much-reduced parameter sets.

## 1 INTRODUCTION

Large language models (LLMs) are capable of completing various tasks (Kasneci et al., 2023; Wu et al., 2023; Thirunavukarasu et al., 2023; Wei et al., 2022). The transformer model (Vaswani et al., 2017), the key behind current prevailing LLMs, is known to have strong in-context learning (ICL) capabilities, and concrete ICL results for transformers have been established for some simple tasks (Garg et al., 2022; Von Oswald et al., 2023; Bai et al., 2024; Ahn et al., 2024). Despite these results, the mechanism for transformers to learn in context is still not fully understood, especially when the scenario is complex or the sequences have memories. Toward this goal, several recent works studied how transformers can learn fixed-order Markov chains (FOMCs) either in training or in-context (Makkuva et al., 2024; Edelman et al., 2024), where insightful observations and theoretical results were obtained. The FOMC is however a poor match for natural languages, for which variable-order Markov chains (VOMCs), also known as context tree (CT) models (Rissanen, 1983; Willems et al., 1995), are often viewed as a more suitable model (Begleiter et al., 2004).

In another related line of research on LLMs, several recent works made explicit the close connection between language models and data compression (Valmeekam et al., 2023; Delétang et al., 2023), the latter of which was aptly titled "Language Modeling Is Compression". Indeed, despite their difference in the eventual applications, the key driver behind both is an accurate (auto-regressive) probability distribution estimator. We adopt this perspective and use the compression rates in a fixed

context window as our main evaluation metric. This allows us to use several well-known compression algorithms, such as prediction by partial matching (PPM) (Cleary & Witten, 1984) and context weighting algorithms (CTW) (Willems et al., 1995), as baselines. In particular, the CTW algorithm is Bayesian optimal under certain priors, which gives us a fundamental lower bound in such settings.

In this work, we consider the ICL of VOMCs from the data compression perspective and refer to this task as ICL-VOMC. The VOMC sources have finite memory, however, each new symbol may depend on different numbers of previous symbols, i.e., the length of the memory may vary. We emphasize that ICL-VOMC is considerably more complex than ICL of FOMCs. A naive strategy would be to view a VOMC as an FOMC of the largest possible order, which however leads to highly inefficient ICL, since the in-context samples are not utilized well. The PPM algorithms can be viewed as an approximate surrogate with this naive approach in the small alphabet setting we consider.

We first train a set of shallow transformers of various numbers of layers for VOMCs of various maximum orders, and empirically observe a few phenomena: 1) Transformers can indeed learn to compress VOMC in context, while PPM suffers significantly; 2) The performance of transformers is not very sensitive to the number of layers, and even a two-layer transformer can learn in-context quite well, tracking the CTW performance closely; and 3) Transformers trained and tested on non-CTW-priors can significantly outperform the CTW algorithm. To explain these phenomena, we carefully analyze the attention maps of the transformers, and discover two attention mechanisms, which form suffixes and perform suffix matching, respectively. Equipped with these mechanisms, we answer the question of whether transformers have the modeling capability to mimic the CTW algorithm. One difficulty is that the CTW algorithm has a recursive structure, which is not directly compatible with the transformer architecture. We first propose an alternative algorithm representation, based on which a transformer construction with $D + 2$ layers is proposed, that can mimic CTW accurately for CTs of maximum order $D$. This establishes a fundamental capability of transformers for ICL-VOMC. The alternative representation enjoys an intuitive interpretation as blending probability estimates along a path on the context tree.

The CTW algorithm relies heavily on counting the occurrences of suffixes of varying lengths to determine the blending coefficients, and a significant component of our transformer construction is for such counting in the second layer. Given its importance, we postulate that 2-layer transformers perform well because such information already allows close-to-optimal blending. We propose such a simple 2-layer transformer, by providing one feed-forward (FF) layer with the probability estimates and the corresponding counts directly. The FF layer's role then largely reduces to approximating the proper blending coefficients. We implement several synthetic transformer layers and show that the hybrid transformers can mostly match the ICL performance of transformers. This construction provides an explanation for 2-layer transformers to perform well in ICL-VOMC, and also for the superior performance on non-CTW-priors. More interestingly, some of these synthetic transformers can perform even better despite the much-reduced parameter sets.

Among existing works, the mostly closely related is Edelman et al. (2024), which studied two-layer transformers and investigated the training behavior from which the transformer obtains its ICL capability of FOMC, leading to conclusive results on a simple binary Markov model. As mentioned before, we focus on variable-order Markov chains (VOMC) and on the ICL behavior in the context window, since we believe it is not only important to be able to learn in context eventually, it is equally important to learn in context quickly. Appendix A gives a more detailed discussion of related works.

**Main Contributions:** We believe that ours is the first study of ICL for VOMC, and the contributions are summarized as follows. (i) We demonstrate that transformers can indeed (numerically) learn to compress VOMC in-context, close to optimal CTW algorithm for appropriate CTW-prior. (ii) We show that transformers can outperform CTW when trained and tested on non-CTW-priors. (iii) We give an explicit $D + 2$-layer transformer construction to imitate CTW establishing its capabilities, based on a novel Bayesian optimal next token prediction representation, which can be of independent interest. (iv) Our construction allows us to investigate the relative insensitivity to the number of layers, of the transformer performance, *i.e.,* even 2-layer transformers perform well. It also gives partial explanations of the ICL capabilities of transformers on VOMC.

**Notation:** Scalars, symbols, and strings are denoted by italic letters like $n$, $N$, $x$, and $s$. Denote by string $x_i^j := (x_i, x_{i+1}, \ldots, x_j)$ as a sequence of symbols. Define () or $x_i^j$ with $i > j$ as an empty string. Vectors and matrices are in bold like $\mathbf{x}$, $\mathbf{H}$, and sets in calligraphic like $\mathcal{A}$ with cardinally $|\mathcal{A}|$.

## 2 PRELIMINARIES

### 2.1 THE TRANSFORMER MODEL

Transformer interacts with sequential data, e.g., $x_1^N = (x_1, \ldots, x_N)$, where token $x_i$ is a symbol from an alphabet (a.k.a. vocabulary) $\mathcal{A}$ with $A = |\mathcal{A}|$. Each token $x_i$ is embedded into $\mathbf{h}_i^{(1)} \in \mathbb{R}^E$ by integrating the information of its value $x_i$ and position $i$, where $E$ is the embedding dimension.

We introduce an $L$-layer decoder-only transformer model. Each layer of the transformer takes matrix $\mathbf{H}^{(\ell)} = [\mathbf{h}_1^{(\ell)}, \mathbf{h}_2^{(\ell)}, \ldots, \mathbf{h}_N^{(\ell)}]$, where $\mathbf{h}_i^{(\ell)} \in \mathbb{R}^E$, as its input and applies the multi-head attention (MHA) layer operation and the feed-forward layer operation, and the output of the layer is the input to the next layer, denoted as $\mathbf{H}^{(\ell+1)}$. The decoder-only multi-head attention layer with $M^{(\ell)}$ heads is

$$\mathbf{a}_i^{(\ell)} = \text{MHA}\left(\mathbf{h_i}, \mathbf{H}; \{W_{O,m}^{(\ell)}, W_{Q,m}^{(\ell)}, W_{K,m}^{(\ell)}, W_{V,m}^{(\ell)}\}_{m=1}^{M^{(\ell)}}\right) \triangleq W_O^{(\ell)}\left[\mathbf{b}_{1,i}^{(\ell)}; \mathbf{b}_{2,i}^{(\ell)}; \ldots; \mathbf{b}_{M^{(\ell)},i}^{(\ell)}\right], \quad (1)$$

where $\{W_{Q,m}^{(\ell)}, W_{K,m}^{(\ell)}, W_{V,m}^{(\ell)}\}_{m=1}^{M^{(\ell)}}$ are the $E^{(\ell)} \times E$ query matrices, key matrices, and value matrices[1] at the $\ell$-th layer and $m$ is the index of the attention head, respectively, $W_O^{(\ell)}$ is the $E \times (M^{(\ell)} E^{(\ell)})$ output mapping matrix, and $\mathbf{b}_m^{(\ell)}$ is the output of the $m$-th attention head at this layer defined as

$$\mathbf{b}_{m,i}^{(\ell)} = (W_{V,m}^{(\ell)}[\mathbf{h}_1^{(\ell)}, \mathbf{h}_2^{(\ell)}, \ldots, \mathbf{h}_i^{(\ell)}]) \cdot \text{softmax}((W_{K,m}^{(\ell)}[\mathbf{h}_1^{(\ell)}, \mathbf{h}_2^{(\ell)}, \ldots, \mathbf{h}_i^{(\ell)}])^\top (W_{Q,m}^{(\ell)} \mathbf{h}_i^{(\ell)})), \quad (2)$$

where we used ";" to indicate vertical matrix concatenation and "," to indicate horizontal matrix concatenation. The attention layer has a residual connection, and the attention output together with the residual connection also goes through a feed-forward layer with a residual connection

$$\mathbf{h}_i^{(\ell+1)} = \text{FF}(\mathbf{a}_i^{(\ell)}; W_1^{(\ell)}, W_2^{(\ell)}) = W_1^{(\ell)}\sigma(W_2^{(\ell)}(\mathbf{a}_i^{(\ell)} + \mathbf{h}_i^{(\ell)})) + (\mathbf{a}_i^{(\ell)} + \mathbf{h}_i^{(\ell)}), \quad (3)$$

where $\sigma$ is a non-linear activation function (e.g., ReLU or sigmoid). The output of the last ($L$-th) transformer layer $\mathbf{H}^{(L+1)}$ goes through a linear then softmax unit to predict the probability of generating the next symbol in vocabulary $\mathcal{A}$ based on the past observations:

$$\hat{\mathbf{p}}_{i+1} = \text{softmax}(W_O^{(L+1)}\mathbf{h}_i^{(L+1)}) \in \Delta_A, \quad i = 1, \ldots, N-1, \quad (4)$$

where $\Delta_A$ is the probability simplex on $\mathcal{A}$. The model is illustrated by figures in Appendix B.1.

Transformers are (pre-)trained to predict next token by minimizing the log-loss (cross-entropy loss) $\mathbb{E}_{x_1^N}[\sum_{i=1}^{N-1} \mathbf{x}_{i+1}^\top \log(1/\hat{\mathbf{p}}_{i+1})]$, where $\mathbf{x}_i \in \mathbb{R}^A$ is the one-hot encoding of $x_i$ and sequence $x_1^N$ is sampled from some population of sources, e.g., sequences can be articles written by different authors and thus following different statistical dynamics.

### 2.2 IN-CONTEXT LEARNING AS BAYESIAN UNIVERSAL CODING

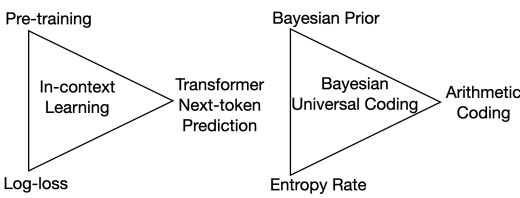

Figure 1: ICL v.s. Bayesian Universal Coding

ICL has a natural connection to compression in information theory (Delétang et al., 2023). Information theory proves that a stationary data source can be compressed losslessly at a rate no less than its entropy rate (Cover & Thomas, 1991). A well-known compression algorithm is arithmetic coding (Rissanen & Langdon, 1979; Pasco, 1976; Rissannen, 1976), which requires an estimated *probability distribution $\hat{P}$ for the next source symbol* to compress a data source that follows the true distribution $P$. The precise compression mechanism is somewhat complicated, and theoretical guarantees vary depending on the structure of the underlying data sources. Nevertheless, it suffices for us to view it as a black box that compresses a symbol $x$ with approximately $\ln(1/\hat{P}(x))$ nats, resulting in an accumulated rate roughly equal to the cross-entropy between $\hat{P}$ and $P$; when $\hat{P} = P$, this reduces to the entropy rate. Note that the entropy rate is a lower bound for the asymptotic averaged log-loss of the transformers.

The learning aspect of ICL closely resembles universal compression (Rissanen, 1983). Naïvely speaking, the latter aims to adaptively compress the source given its context, essentially approaching

---

[1]In practice, embedding dimension $E$ is divisible by the number of heads $M^{(\ell)}$ and $E = M^{(\ell)} E^{(\ell)}$.

its entropy rate without having direct access to the underlying dynamics, but learning it in an in-context fashion. We illustrate the close analogy between ICL and Bayesian universal coding in Fig. 1, where pretraining of learning is equivalent to learning the Bayesian prior of the sources, and the next token prediction can be viewed as a part of arithmetic coding (AC), both measured by log-loss.

## 2.3 CONTEXT TREE MODELS (VARIABLE-ORDER MARKOV CHAINS)

Variable-order Markov chains (VOMCs), also known as context tree (CT) models, have been studied extensively in the data compression literature (Rissanen, 1983; Willems et al., 1995; Begleiter et al., 2004). String $s = (x_{1-l}, x_{2-l}, \ldots, x_0)$ is a suffix of the string $s' = (x'_{1-l'}, x'_{2-l'}, \ldots, x'_0)$, if $0 \le l \le l'$ and $x_{-i} = x'_{-i}$ for $i = 0, 1, \ldots, l-1$; e.g., $(a, b, c, b)$ is suffix of $(a, c, a, a, b, c, b)$. Note that the strings above have non-positive indices.

The statistical behavior of a finite memory CT source is specified by a suffix set $\mathcal{S}$ and the associated next token probability distributions. The suffix set is a collection of strings $s(k)$, $k = 1, 2, \ldots, |\mathcal{S}|$, which needs to be proper and complete: The set is proper if no string in $\mathcal{S}$ is a suffix of any other string; it is complete if each semi-infinite sequence $(\ldots, x_{n-1}, x_n)$ has a unique suffix that belongs to $\mathcal{S}$, denoted as $\beta_{\mathcal{S}}(\ldots, x_{n-1}, x_n)$. Associated with each suffix $s \in \mathcal{S}$, there is a probability mass function $p_s \in \Delta_{\mathcal{A}}$. A CT has maximum order $D$ if any suffix in $\mathcal{S}$ has a length at most $D$. Given a semi-infinite sequence $(\ldots, x_{n-1}, x_n)$, the next symbol $x_{n+1}$ is generated randomly according to the distribution $p_{\beta_{\mathcal{S}}(\ldots, x_{n-1}, x_n)}$. An example CT is in Fig. 2. A tree structure appears since for any valid suffix set $\mathcal{S}$, there exists a unique

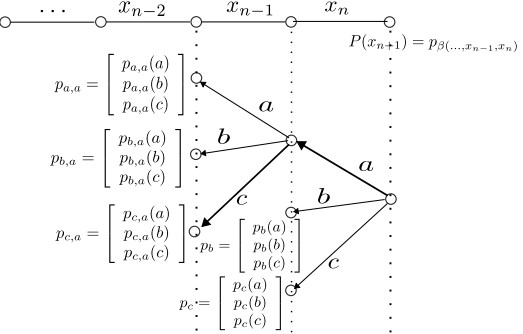

Figure 2: A CT in the alphabet $\mathcal{A} = \{a, b, c\}$ with suffix set $\mathcal{S} = \{(b), (c), (a, a), (b, a), (c, a)\}$ and the associated probability distributions. If $(\ldots, x_{n-1}, x_n) = (\ldots, c, a)$, then the probability distribution for the next symbol $x_{n+1}$ is $p_{c,a}$.

tree $T$ with $\mathcal{S}$ being its leaves $\mathcal{L}(T)$. A CT can thus be equivalently represented by $(T, \{p_s\}_{s \in \mathcal{L}(T)})$.

## 2.4 BAYESIAN CONTEXT TREE WEIGHTING COMPRESSION ALGORITHM

Once the underlying CT is estimated accurately, AC can be used to compress the sequence efficiently. The difficulty in estimating a context tree is in finding both of its components: the tree structure itself, and the probability distribution associated with each leaf node. The likelihood of a sequence $x_1^i$ given $x_{1-D}^0$ for a CT with parameter $(T, \{p_s\}_{s \in \mathcal{L}(T)})$ is

$$P_{T, \{p_s\}}(x_1^i | x_{1-D}^0) = \prod_{j=1}^{i} p_{\beta_{\mathcal{L}(T)}(x_{j-D}, \ldots, x_{j-1})}(x_j) = \prod_{s \in \mathcal{L}(T)} \prod_{a \in \mathcal{A}} p_s(a)^{\mathbf{n}_{i,s}(a)},$$

where $\mathbf{n}_{i,s}$ is the *counting vector* associated with suffix $s$ that

$$\mathbf{n}_{i,s}(a) := \text{number of times symbol } a \in \mathcal{A} \text{ follows suffix } s \text{ in sequence } (x_1, \ldots, x_i). \quad (5)$$

Leveraging the multiplicative nature of the likelihood function, Willems et al. (1995) proposed the context tree weighting (CTW) algorithm for CT sources with maximum order $D$ based on the minimum description length principle. CTW estimates the probability of the sequence $x_1^n$ by the auxiliary parameters $p_{n,s}^e, p_{n,s}^w$'s as follows.

1. For each $s \in \mathcal{A}^*$ with $|s| \le D$, compute $p_{n,s}^e = \frac{\Gamma(\sum_{a \in \mathcal{A}} \boldsymbol{\alpha}(a))}{\Gamma(\sum_{a \in \mathcal{A}} (\mathbf{n}_s(a) + \boldsymbol{\alpha}(a)))} \prod_{q \in \mathcal{A}} \frac{\Gamma(\mathbf{n}_s(a) + \boldsymbol{\alpha}(a))}{\Gamma(\boldsymbol{\alpha}(a))}$, where $\mathbf{n}_s$ is the counting vector $\mathbf{n}_{i,s}$ with $i = n$, $\Gamma(\cdot)$ is the Gamma function, and $\boldsymbol{\alpha}$ is a prior-related vector that will be specified later.

2. From nodes in the $D$-th level to the 0-th level (i.e., root), iteratively compute

$$p_{n,s}^w := \begin{cases} p_{n,s}^e, & \text{if } |s| = D, \\ \lambda p_{n,s}^e + (1 - \lambda) \prod_{q \in \mathcal{A}} p_{n,qs}^w, & \text{otherwise,} \end{cases} \quad (6)$$

where $qs$ is the string by appending symbol $q \in \mathcal{A}$ before the suffix $s$.

Kontoyiannis et al. (2022) took the Bayesian view towards this procedure, and showed that the probability $p_{n,()}^w$ at the root has a clear Bayesian interpretation under a CTW prior. CTW prior $\pi_{\text{CTW}}$ is a Bayesian CT prior over the trees in $\mathcal{T}(D) := \{$full $A$-ary tree with depth at most $D\}$ and the transition distributions $p_s \in \Delta_{\mathcal{A}}$. Specifically, $\pi_{\text{CTW}}(T, (p_s)_{s \in \mathcal{L}(T)}) = \pi_D(T) \prod_{s \in \mathcal{L}(T)} \pi_p(p_s)$, where $\pi_D(\cdot)$ represents a bounded branching process that each node at a level lower than $D$ stops branching with probability $\lambda$ or branches to $|\mathcal{A}|$ children with probability $(1-\lambda)$; and $\pi_p(p_s)$ satisfies a Dirichlet distribution. Mathematically,

$$\pi_D(T) = (1-\lambda)^{(|\mathcal{L}(T)|-1)/(A-1)} \lambda^{|\mathcal{L}(T)|-|\mathcal{L}_D(T)|}, \quad \pi_p(p_s) = \text{Dir}(p_s; \{\boldsymbol{\alpha}(a)\}_{a \in \mathcal{A}}),$$

where $\mathcal{L}_D(T)$ is the leaves of $T$ with depth $D$ and $\{\boldsymbol{\alpha}(a)\}_{a \in \mathcal{A}}$ are the Dirichlet parameters. A typical choice is $\boldsymbol{\alpha}(a) = 0.5$ for each $a \in \mathcal{A}$ corresponding to the Jeffreys prior.

**Theorem 1.** *(Kontoyiannis et al., 2022, Theorem 3.1) The $p_{n,()}^w$ value at root computed by the CTW procedure equals to the Bayesian predicted probability under prior $\pi_{CTW}$ specified by $(D, \lambda, \boldsymbol{\alpha})$:*

$$p_{n,()}^w = P_{\pi_{CTW}}(x_1^n | x_{1-D}^0) = \sum_{T \in \mathcal{T}(D)} \int P_{T, \{p_s\}}(x_1^n | x_{1-D}^0) \pi(T, \{p_s\}) \Big( \prod_{s \in \mathcal{L}(T)} dp_s \Big).$$

This theorem implies that the CTW procedure exactly matches the Bayesian CTs with prior $\pi$ parameterized by $(D, \lambda, \boldsymbol{\alpha})$ and the probability of sequence $x_1^n$ is $p_{n,()}^w$, i.e., the $p^w$ at the root. AC can be applied via sequentially calculating the predictive next token probability as $P_{\pi_{\text{CTW}}}(x_{i+1} | x_{1-D}^i) = \frac{P_{\pi_{\text{CTW}}}(x_1^{i+1} | x_{1-D}^0)}{P_{\pi_{\text{CTW}}}(x_1^i | x_{1-D}^0)}$ and achieves code length at most $\lceil \log_2(1/p_{n,()}^w) \rceil$ (Willems et al., 1995; 1997).

Besides CTW, the prediction by partial matching algorithm (PPM) (Cleary & Witten, 1984) is also well known for its good performance in practice (Begleiter et al., 2004). PPM takes the maximal order $D_{\text{ppm}}$ as a parameter and blends CTs by utilizing an escape symbol. The key idea is when a suffix is less observed, the escape symbol is encoded, indicating a shorter suffix needs to be used to predict the next token probability. More details of PPM are given in Appendix B.2.

## 3 Transformers Learn In-context of VOMCs

### 3.1 The ICL-VOMC Task, Transformer Training, and Testing

We choose ternary alphabet $|\mathcal{A}| = 3$, and pretrain a transformer of context window size $N$ on data sequences of length-$N$ generated using CTs randomly sampled from a CTW prior $\pi_{\text{CTW}}$ parameterized by $\boldsymbol{\alpha} = 0.5$, $\lambda = 0.15$ and a fixed maximum tree depth $D$, illustrated in Fig. 3. The training loss is the canonical next-token prediction cross-entropy loss. During the inference, given a source sequence of length-$N$ generated from an unknown VOMC with an order at most $D$, can the transformer compress this sequence efficiently, i.e., at a compression rate close to the optimal rate?

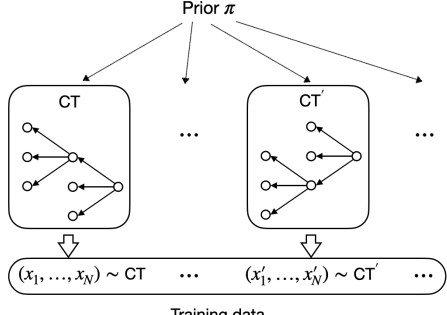

Figure 3: Training data collection

### 3.2 Transformers Can Learn VOMC In-Context

In Fig. 4, we show the performance comparisons between trained transformers with various numbers of layers, and the reference PPM and CTW algorithms for $N = 1536$ and CT maximum order $D = 5$. The transformers have 8 attention heads with embedding dimension $E = 128$; in our settings, we found the performance of the transformers is not constrained by these parameters. In Table 1, we further provide the average compression rates over the whole context window for CTs of different orders; we refer to the transformers as TF, and TF-$L$ refers to having $L$ layers. For CTs with lower order, the transformer embedding dimension $E$ is set at 64 instead of 128.

A few observations are immediate.

**1)** The CTW algorithm is Bayesian optimal in this setting, and it provides a lower bound for other methods as expected.

|           | TF-1   | TF-2   | TF-3   | TF-4   | TF-5   | TF-6   | CTW    |
|-----------|--------|--------|--------|--------|--------|--------|--------|
| CTs $D = 3$ | 0.9368 | 0.7297 | 0.7265 | 0.7220 | 0.7245 | 0.7258 | 0.7165 |
| CTs $D = 4$ | 0.9667 | 0.7831 | 0.7818 | 0.7759 | 0.7791 | 0.7774 | 0.7603 |
| CTs $D = 5$ | 0.9661 | 0.7569 | 0.7490 | 0.7440 | 0.7437 | 0.7438 | 0.7400 |

Table 1: Average compression rates in the context window by transformers and CTW, where the CTs are sampled from the CTW-prior. The context window and embedding dimension for CTs of $D = 5$ are $N = 1536$ and $E = 128$, while for others it are $N = 512$ and $E = 64$.

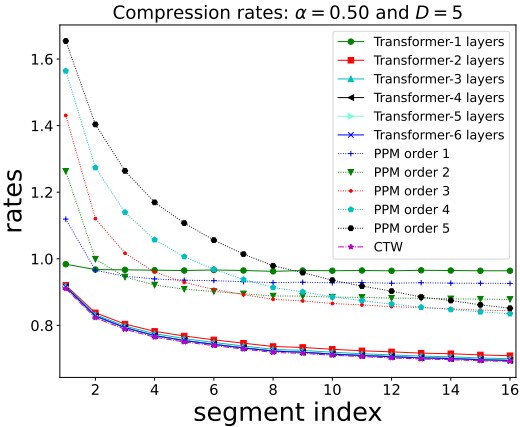

Figure 4: Transformer, PPM, CTW

**2)** The PPM algorithms perform poorly in this setting, which is expected since they essentially reduce to FOMC estimators at the assumed maximum order $D_{\mathrm{ppm}}$, for the small alphabet setting we consider. Small $D_{\mathrm{ppm}}$ leads to oversimplification bias to the model and thus performs poorly. However, even when $D_{\mathrm{ppm}}$ in PPM is sufficiently large, i.e., $D_{\mathrm{ppm}} \geq D$, it is a highly inefficient estimator for those contexts at lower orders in the CT. Therefore we can view PPM as a reference method that does not adapt to the variable orders efficiently. The particular poor performance of the PPM algorithm at the start of the sequence is due to the escape symbol encoding, however, toward the end of the sequence, PPM starts to improve if $D_{\mathrm{ppm}} \geq D$.

**3)** Most interestingly, almost all trained transformers, except that with a single layer, track the performance of the CTW algorithm fairly closely. The overall performance does improve as the number of layers increases in general; see Table 1 for numerical comparisons. Nevertheless, the improvements with increased numbers of layers are relatively small and appear to saturate at four layers. Particularly, even transformers with two layers appear to learn in context quite well.

In the next section, we provide theoretical and empirical explanations for these observations.

### 3.3 TRANSFORMERS VS. CTW UNDER NON-CTW-PRIORS

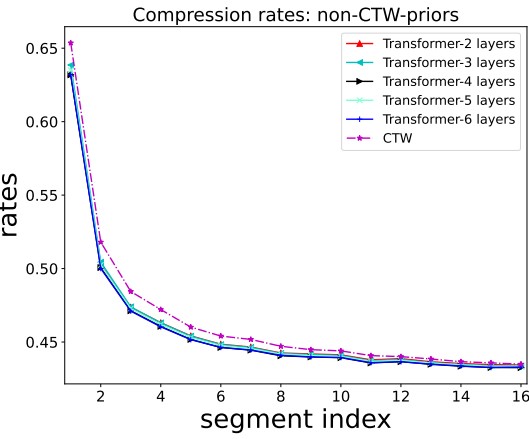

Figure 5: Transformers vs. CTW

The CTW algorithm is known to be Bayesian optimal when the CTs are generated from a CTW-prior. When the CTs do not follow those priors, can learning-based transformers perform better than CTWs? We empirically observe that in such settings, transformers indeed have advantages. The training data are generated by using CTs of different maximum orders, where the orders are chosen uniformly at random between 1 and 3. Moreover, the probability vector is not generated from the Dirichet prior, but from a distribution that for each CT leaf, randomly assigns one of the elements in the alphabet to have zero probability. We test on sequences generated from CTs produced from the same distribution as in the training setting. We assume the CTW takes the default (non-informative prior) parameters of $\boldsymbol{\alpha}(a) = 0.5$, and the same tree branch stopping parameter $\lambda = 0.15$ as taken in the testing sequence CTs.

As can be observed in Fig. 5, the CTW algorithm is no longer optimal, and trained transformers can perform considerably better. In fact, even transformers with 2 layers can outperform the CTW

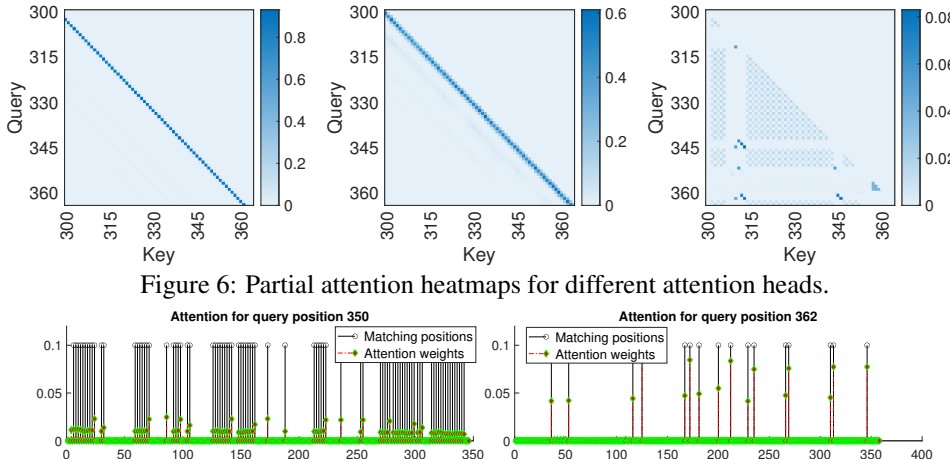

Figure 6: Partial attention heatmaps for different attention heads.

Figure 7: Suffix locations and attention weights in the second type of pattern at two query positions.

algorithm in this setting, and more layers usually lead to further improved performance, albeit the improvement is less significant.

# 4 THEORETICAL INTERPRETATIONS AND EMPIRICAL EVIDENCES

To understand why and how the trained transformers perform comparable to CTW, we first analyze their attention maps, which reveal interesting patterns and behaviors. We then propose a transformer-friendly representation of the Bayesian optimal next token prediction by CTW, and motivated by the attention map observations, we provide transformer constructions to interpret its ICL capabilities and capacities. We next implement these synthetic transformer layers, and show that the hybrid transformers can match the ICL performance of the original version of transformers, which serves as evidence for the proposed constructions.

## 4.1 ANALYSIS OF ATTENTION MAPS

We analyzed the attention maps of the trained transformers where a few distinguished patterns emerge. One pattern is solely relative-position dependent. In the left two panels of Fig. 7, we observe off-diagonal stripes for these two attention heads, which are a few positions below the main diagonal. They can be a single off-diagonal or a collection of several off-diagonals. This indicates that the query position is attending positions at a few fixed but close distances ahead of itself. This pattern usually appears in the first or second layers of the transformers. Combining with the suffix structure in compression algorithms such as CTW, such an attention pattern suggests the suffix is being copied into the current query position for subsequent processing. The off-diagonal stripes may have a width greater than 1, as shown in the second panel, which can be viewed as copying a mixture of the tokens in the suffix, suggesting the flexibility of transformers in forming certain "soft" suffixes.

Another pattern, shown in the third panel has more sophisticated spotty patterns, and the attention appears to depend more explicitly on the current token features instead of the position alone, and they usually appear in the second layer or above in the transformers. Taking query positions 350 and 362 for the attention head shown in the third panel of Fig. 6, we plot in Fig. 7 the positions in the data sequence that match their suffixes of length-3 using the stem plots with a black circle on top, and the attention values as the red stems with the diamonds on top. It can be seen that the positions match perfectly, though the attention weights have some variations among them. The left panel has more matching locations, due to the inherent Markov chain structure. This attention pattern suggests that it is collecting information for those positions with the matched suffix of a fixed length. Several attention heads present similar patterns but with different suffix lengths.

## 4.2 CAPABILITY AND CAPACITY OF TRANSFORMER VIA CONSTRUCTION

Motivated by the observations of the attention map patterns, we connect the performance of transformers to CTW via construction.

#### 4.2.1 A REPRESENTATION OF CTW OPTIMAL NEXT TOKEN PREDICTION

Given a sequence $x_1^n$ generated according to a $\text{CT}(T, \{p_s\})$ sampled from the CTW-prior $\pi_{\text{CTW}}$ parameterized by $(D, \lambda, \boldsymbol{\alpha})$, we propose a novel representation for computing the predictive probability $P_{\pi_{\text{CTW}}}(x_{n+1}|x_{1-D}^n)$ in the following theorem, which predicts $x_{n+1}$ based on the weighted blending of the next token prediction probability vectors corresponding to each potential suffix $s_{n,l} := x_{n-l+1}^n$ of length $l = 0, 1, \ldots, D$. The proof of Theorem 2 is in Appendix D.1.

**Theorem 2.** *The predicted probability can be computed as*

$$P_{\pi_{\text{CTW}}}(x_{n+1}|x_1^n) = \sum_{l=0,\ldots,D} \omega_{n,l} \cdot \mathbf{p}_{n,s_{n,l}}(x_{n+1}), \tag{7}$$

*where* $\mathbf{p}_{n,s_{n,l}}(a) = \frac{\boldsymbol{\alpha}(a)+\mathbf{n}_{n,s_{n,l}}(a)}{\sum_q (\boldsymbol{\alpha}(q)+\mathbf{n}_{n,s_{n,l}}(q))}$*; and* $\omega_{n,\cdot} \in \Delta_{D+1}$ *with* $\ln(\omega_{n,l})-\ln(\omega_{n,l-1}) = \ln(1-\lambda)-\mathbb{I}_{l=D}\ln(\lambda)+\ell_{n,s_{n,l}}^e-\ell_{n,s_{n,l-1}}^e+\sum_{q\in\mathcal{A}}\ell_{n,qs_{n,l-1}}^w-\ell_{n,s_{n,l}}^w$ *for* $l = 1, \ldots, D$*, where* $\ell_{n,s}^e = \ln(p_{n,s}^e)$*,* $\ell_{n,s}^w = \ln(p_{n,s}^w)$ *, and* $\mathbb{I}_{(\cdot)}$ *is the indicator function.*

As illustrated in Fig. 8, each suffix $s_{n,l}$, e.g., $s_{n,0} = ()$, $s_{n,2} = ba$, can potentially be the true suffix of the underlying CT dynamics, i.e., $s_{n,l} \in \mathcal{L}(T)$; and $\mathbf{p}_{n,s_{n,l}}$ is in fact the Bayesian optimal next token prediction given $s_{n,l} \in \mathcal{L}(T)$. The blending weights $\omega_{n,l}$ assign credibility that $s_{n,l}$ is the true suffix. As shown in Theorem 2, the weights are based on stopping probability $\lambda$, the information in the potential suffix path such as $p_{s_{n,s_{n,l}}}^e$ as well as the information from their siblings $p_{n,qs_{n,l-1}}^w$ (siblings and their sub-trees are in triangles in Fig. 8). The information of counting vector $\mathbf{n}_{n,s}$ plays a vital role since $\mathbf{p}_{n,s}, p_{n,s}^e$, e.t.c. are all functions of $\mathbf{n}_{n,s}$.

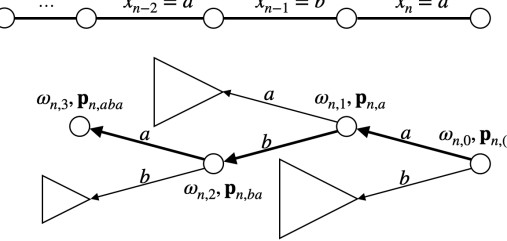

Optimal next token prediction: Weighted averaging along path
$P_\pi(\cdot|x_1^n) = \omega_{n,0}\mathbf{p}_{n,()}(\cdot) + \omega_{n,1}\mathbf{p}_{n,a}(\cdot) + \omega_{n,2}\mathbf{p}_{n,ba}(\cdot) + \omega_{n,3}\mathbf{p}_{n,aba}(\cdot)$

Figure 8: Illustration of Theorem 2

#### 4.2.2 TRANSFORMER CONSTRUCTION: APPROXIMATING CTW

We provide a construction of $(2 + D)$-layer transformer with sufficient representation power in the FF layer that can essentially approximate CTW, which demonstrates the capacity of transformers. The first two layers are motivated by the attention map patterns observed in Section 4.1, which we show their capabilities of capturing the important counting vector statistics suggested by Theorem 2. The last $D$ layers are induction layers imitating the CTW procedure.

We consider the initial embedding is one-hot, with additional scratch pad elements initialized as zeros and a positional embedding, i.e., $\mathbf{h}_i^{(1)} = (\mathbf{x}_i; \mathbf{0}; \mathbf{pos}_i)$ where we used $\mathbf{x}_i \in \mathbb{R}^A$ to denote the one-hot (column vector) embedding of $x_i$ , $\mathbf{pos}_i = (1, \cos(i\pi/N), \sin(i\pi/N))^\top$ is a sinusoidal positional embedding, and the remaining $(E - A - 3)$ elements being zero. The parameter $E$ will be specified later, and let us assume it is sufficiently large at this point.

We begin with the first layer, which is referred to as a finite-memory context-extension layer.

**Theorem 3.** *There is an $M$-headed transformer layer that can perform finite-memory context-extension, defined by the following output, with the initial one-hot embedded input $\mathbf{H}^{(1)}$:*

$$\mathbf{h}_i^{(2)} = (\mathbf{s}_{i,M+1}; \mathbf{0}; \mathbf{pos}_i), \tag{8}$$

*where* $\mathbf{s}_{i,M+1} = (\mathbf{x}_i; \ldots; \mathbf{x}_{i-M})$ *is the vector version of the $M$-length suffix $s_{i,M+1} = x_{i-M}^i$.*

This layer essentially copies $M$ past embedded symbols to the current position $i$, and stacks them below the current symbol $\mathbf{x}_i$. This operation utilizes the positional encoding $\mathbf{pos}_i$ via rotation and matching the corresponding positions. The detailed construction and proof is in Appendix D.2.1.

The second layer is referred to as the statistics collection layer, which takes a sequence of vectors $\mathbf{h}_i^{(2)}$, $i = 1, \ldots, N$, defined in (8) as its input. To rigorously specify the function of this layer, we define the $k$-gram (forward) statistics vector $\mathbf{g}_{i,s}$ with $|s| = k - 1$, which in plain words, is the empirical

probability distribution of the next token associated with the suffix $s$ for sequence $x_1^i$. Similarly, we define the $k$-gram backward statistics vector $\mathbf{g}_{i-1,s}^{\leftarrow}$, which is the empirical probability distribution of the previous token associated with the suffix $s$ for $x_1^{i-1}$. Mathematically, for a suffix $s$ and position $i$,

$$\mathbf{g}_{i,s}(a) = \frac{\mathbf{n}_{i,s}(a)}{\sum_{q \in \mathcal{A}} \mathbf{n}_{i,s}(q)}, \qquad \mathbf{g}_{i-1,s}^{\leftarrow}(a) = \frac{\sum_{q \in \mathcal{A}} \mathbf{n}_{i,as}(q)}{\sum_{q \in \mathcal{A}} \mathbf{n}_{i,s}(q)}, \qquad \forall a \in \mathcal{A}, \tag{9}$$

where $\mathbf{n}_{i,s}$ is the counting vector defined in (5), and $\sum_{q \in \mathcal{A}} \mathbf{n}_{i,s}(q)$ is the number of appears of the string $s$ in the sequence $x_1^{i-1}$. For both $\mathbf{g}_{i,s}$ and $\mathbf{g}_{i-1,s}^{\leftarrow}$, if the suffix $s$ has not appeared in data $x_1^{i-1}$, it can be initialized arbitrarily as a vector in the probability simplex.

**Theorem 4.** *There is an $M'$-head attention layer, where $M' \leq M + 1$, that can perform statistics collection, defined by the following output, with $\mathbf{H}^{(2)}$ in (8) as its input:*

$$\mathbf{a}_i^{(2)} = (\mathbf{s}_{i,M+1}; \mathbf{g}_{i,M'}; \mathbf{g}_{i-1,M'}^{\leftarrow}; \mathbf{0}; \mathbf{pos}_i), \tag{10}$$

*where $\mathbf{g}_{i,M'} := (\mathbf{g}_{i,s_{i,0}}; \ldots; \mathbf{g}_{i,s_{i,M'-1}})$ and $\mathbf{g}_{i-1,M'}^{\leftarrow} = (\mathbf{g}_{i-1,s_{i,0}}^{\leftarrow}; \ldots; \mathbf{g}_{i-1,s_{i,M'-1}}^{\leftarrow})$.*

This functional layer essentially collects $k$-gram statistics for various lengths of $k = 1, 2, \ldots, M'$. For example, when $k = 3$, it collects the normalized frequency associated with the suffix $(x_{n-1}, x_n)$.

For ICL of FOMCs, two-layer transformers collecting forward statistics $\mathbf{g}_{i,M'}$ with $M' = D + 1$ is sufficient (Edelman et al., 2024). However, for the ICL-VOMC task, the underlying CT structure is unknown, therefore, collecting such simple statistics is no longer sufficient. As indicated in Theorem 2, the information of counting statistics $\mathbf{n}_{i,s_{i,l}}$ is important to the performance of prediction since the weights heavily depend on $\mathbf{n}_{i,s}(a)$. Yet due to the softmax function of attention layer, only (normalized) probabilistic vector can be obtained instead of the exact count. With the backward statistics $\mathbf{g}_{i,s}^{\leftarrow}$, $\mathbf{n}_{i,s_{i,l}}$ can be derived as $\mathbf{n}_{i,s_{i,l}}(a) = \frac{\mathbf{n}_{i,s_{i,l}}(a)}{\sum_{q \in \mathcal{A}} \mathbf{n}_{i,s_{i,l}}(q)} \frac{\sum_{q \in \mathcal{A}} \mathbf{n}_{i,s_{i,l}}(q)}{\sum_{q \in \mathcal{A}} \mathbf{n}_{i,s_{i,l-1}}(q)} \cdots \frac{\sum_{q \in \mathcal{A}} \mathbf{n}_{i,s_{i,1}}(q)}{\sum_{q \in \mathcal{A}} \mathbf{n}_{i,s_{i,0}}(q)} \left( \sum_{q \in \mathcal{A}} \mathbf{n}_{i,s_{i,0}}(q) \right) = \mathbf{g}_{i,s_{i,l}}(a) \left( \prod_{j=0}^{l-1} \mathbf{g}_{i-1,s_{i,j}}^{\leftarrow}(x_{i-j}) \right) i$, by the information contained in vector $\mathbf{a}_i^{(2)}$.

Taking $M = M' - 1 = D$, after the statistics collection multi-head attention layer, a sufficiently wide FF layer with ReLU activation gives

$$\mathbf{h}_i^{(3)} = (\mathbf{s}_{i,D}; \mathbf{p}_{i,D}; \mathbf{l}_{i,D}^e; \ell_{i,s_{i,D}}^w; \mathbf{0}; \mathbf{pos}_i), \tag{11}$$

where $\mathbf{p}_{i,D} = (\mathbf{p}_{i,s_{i,0}}; \mathbf{p}_{i,s_{i,1}}; \ldots; \mathbf{p}_{i,s_{i,D}})$ and $\mathbf{l}_{i,D}^e = (\ell_{i,s_{i,0}}^e; \ldots; \ell_{i,s_{i,D}}^e)$ in Theorem 2, due to the universal approximation of wide two-layer neural networks (Cybenko, 1989; Hornik et al., 1989).

To fulfill the Bayesian optimal prediction, we introduce the following induction layer that iteratively compute $\ell_{i,s}^w$ for suffix on the valid suffix path and their siblings, and also the weight difference denoted by $\delta_{i,l} := \ln(\omega_{i,l}) - \ln(\omega_{i,l-1})$ for $l = d, D - 1, \ldots, 1$. Specifically, the desired embedding

$$\mathbf{h}_i^{(\ell)} = (\mathbf{s}_{i,M^{(1)}+1}; \mathbf{p}_{i,D}; \mathbf{l}_{i,D}^e; \delta_{i,D}; \delta_{i,D-1}; \ldots; \delta_{i,D-\ell+4}; \ell_{i,s_{i,D+3-\ell}}^w; \mathbf{0}; \mathbf{pos}_i), \tag{12}$$

for $\ell = 3, 4, \ldots, 3 + D$.

**Theorem 5.** *There exists a $A$-head transformer layer that can perform the induction: Takes $\mathbf{H}^{(\ell)}$ in (12) as input and outputs $\mathbf{H}^{(\ell+1)}$. And the final output layer taking $\mathbf{H}^{(D+3)}$ as input can output the $A$-dimensional Bayesian optimal next token prediction vector $P_{\pi_{CTW}}(\cdot|x_{1-D}^n) = \sum_{l=0,\ldots,D} \omega_{n,l} \mathbf{p}_{n,s_{n,l}}$.*

Although transformers with sufficient FF layers can theoretically compute the optimal prediction as CTW, empirically, transformers of $2 + D$ layers perform slightly worse in our experiments. This is likely due to the less-than perfect pretraining optimization and the limited representation capability of finite-width FF layers with ReLU activations. We also note that the proposed transformers construction may not be the only way to mimic CTW, however, we believe the first two layers do capture important universal features. We provide supporting evidences empirically in the sequel.

### 4.3 A Reduced Two-layer Construction

We conduct experiments on the hybrid versions of transformers. Let "TF 0-2" denote the canonical 2-layer transformer; "TF 1-1" denote the transformer consisted of a constructed layer with output

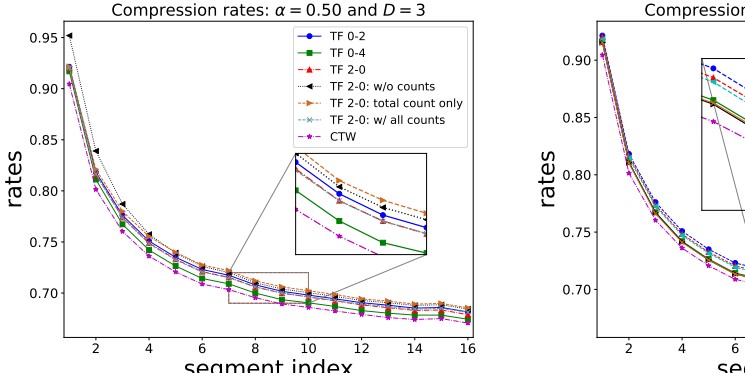

Figure 9: Hybrid Transformers: Effects of accumulative suffix counts and synthetic layers

$\mathbf{h}_i^{(1)}$ (8), and a trainable transformer layer and a output layer taking $\mathbf{H}^{(1)}$ as input; and denote by "TF 2-0" the transformer with 2 constructed layer with output $\mathbf{a}_i^{(2)}$ in (10), followed by a trainable FF layer (the FF layer in the second layer of the transformer) and an output layer.

We first study the key statistics behind the strong performance of two-layer transformers, as shown in the left panel in Fig. 9. Compared to "TF 2-0" which is the constructed layers given previously, the version "TF 2-0 w/o counts" does not contain $\mathbf{g}_{i-1,M'}^{\leftarrow}$ or $\mathbf{pos}_i$ in $\mathbf{a}_i^{(2)}$; the version "TF 2-0 total counts only" does not contain $\mathbf{g}_{i-1,M'}^{\leftarrow}$ in $\mathbf{a}_i^{(2)}$ and $\mathbf{pos}_i$ is replaced by the total count $i$; "TF 2-0 w/ all counts" replaces $\mathbf{g}_{i-1,M'}^{\leftarrow}$ and $\mathbf{pos}_i$ with $\{\mathbf{n}_{n,s_{n,l}}\}_{l=0}^{D}$ and $i$. Even though their performances are rather clustered, we can make the following observations: 1) The performances degrade as more counting information is removed from the representation, and the counting information is clearly very important, 2) The performances of "TF 2-0" and "TF 2-0 w/ all counts" almost match exactly, indicating the main purpose of the backward statistics $\mathbf{g}_{i-1,M'}^{\leftarrow}$ is to extract the counts, and 3) The performance of the original 2-layer transformer is similar to that of the constructed "TF 2-0" and "TF 2-0: w/ all counts" that those without less counting information.

We further study hybrid transformers with the first one or two being the constructed layers. As shown in the right panel of Fig. 9, transformers with 2 total layers and 4 total layers form two clusters, which provides strong evidence that the constructed layers are indeed replacing the first two layers of the original transformers in a functional manner. Moreover, the performances of transformers with a single constructed layer, such as "TF 1-1" and "TF 1-3", are slightly better than those with two constructed layers, such as "TF 2-0" and "TF 2-2", likely due to the flexibility in the remaining trainable transformer layers. Interestingly, for two-layer transformers, the hybrid versions can perform even better than the original transformer "TF 0-2", which we believe is because the latter is having difficulty extracting the exact statistics as those more readily available in the constructed layers.

## 5 CONCLUSION

We considered the in-context learning of transformers for VOMC sources. By drawing a close analogy of ICL and Bayesian universal compression, we leverage the CTW and PPM as baselines. Experimentally, we observe the performances of the trained transformers greatly surpass the performance PPM and are close to that of CTW even with just two layers under CTW priors; moreover, transformers are superior to CTW under non-CTW prior. To understand the mechanism of transformers' ICL ability, we analyzed the attention maps and extracted two likely mechanisms. We then constructed the finite-memory context extension layer, and the statistics collection layer, corresponding to these two mechanisms, respectively. The latter collects both the forward and backward statistics, which are vital as theoretically demonstrated by a novel representation of the CTW optimal next-token prediction. We also provide empirical evidence that the statistics collected by the constructed second layer, in particular the counting statistics, are indeed necessary.

Although we empirically showed transformers can perform ICL-VOMC tasks, and constructed an idealized transformer to mimic the CTW algorithm, it is not clear whether a trained transformer will indeed utilize the upper layer mechanisms. Extending the existing approach (Edelman et al., 2024) to answer this question appears quite difficult, given the complexity of the constructed transformer and the underlying VOMCs; this is part of ongoing investigations.

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

# A  RELATED WORK

There have been many efforts in studying the ICL capabilities of transformers. A significant recent development is the elucidation of the connection to gradient descent, particularly for linear regression tasks (Von Oswald et al., 2023; Akyürek et al., 2022; Dai et al., 2022; Ahn et al., 2024). Li et al. (2023) formulated the ICL problem as a multi-task learning problem and considered ICL for several simple problem settings for which the authors provide risk bounds for ICL of supervised learning algorithms in these problem settings. Kirsch et al. (2022) viewed the ICL problem as a meta-learner and studied the relation between tasks and model sizes. These line of approaches focused on the ICL of supervised learning tasks, such as classification and regression, while this work belongs to another directions of studying ICL for the next token prediction of some unknown underlying dynamics.

Olsson et al. (2022) studied the induction head, i.e., the forming of small $k$-gram attention in LLMs. Reddy (2023) studied the balance between ICL and in-weights learning, and observed the abrupt emergence of the induction head corresponds to the emergence of ICL. The induction head was generalized to the statistical induction head in (Edelman et al., 2024) mainly to study bigrams. We adopted it but further allowed more statistical induction heads for more suffixes to be included together, in the first two layers of the idealized transformer.

There have also been efforts to study transformers and learning of Markov chains. Xie et al. (2021) viewed ICL as a Bayesian inference problem, where a latent concept determines an HHM, and the observations from the HHM can lead to the identification of the hidden concept. They studied the eventual ICL capability, i.e., when the number of in-context examples goes to infinity. Hu et al. (2024) studied the limitations of transformer on learning to perform belief inference for HMMs sources compared to recurrent neural networks. The work in (Bietti et al., 2024) allowed a fixed-order Markov chain to switch to a new deterministic mode, and the authors study the training behavior of the corresponding ICL task with this mode transition. Akyürek et al. (2024) made a comprehensive empirical comparison of various language models on random finite automata, and showed that the transformer performs the best among these models. Makkuva et al. (2024) studied the loss landscape during transformer training on sequences generated from a single fixed-order Markov chain, using a single-layer transformer. Their study does not consider ICL. More recently Rajaraman et al. (2024) considered ICL of FOMCs with single-head transformers, and provided a construction to show that it is possible to use a single attention head to capture longer memory in the sequence. The work most relevant to us is (Edelman et al., 2024), where ICL of a fixed-order Markov chain was considered, and the training behavior was studied both empirically and theoretically, and the forming of induction heads in a two-layer network was demonstrated. All these existing work assumed fixed-order Markov models or fixed-order HHMs, usually with orders kept at 1 or 2; moreover, they almost all focus on the emergence of the induction heads during training or the training landscape. Our study is different firstly in the variable-order nature of the Markov models, and secondly the focus on the on-time ICL performance instead of the training landscape and behavior.

Lossless data compression has a long history, with many different algorithms being developed over the years. The most popular general-purpose compression algorithms are perhaps the Lempel-Ziv compression algorithms (Ziv & Lempel, 1977; 1978) and their variants, which belong to dictionary-based compression algorithms. These algorithms do not explicitly maintain any probabilistic models, and their efficiency comes from maintaining an efficiency dictionary of sequences that have been seen before, and to be matched with future sequences. More powerful compression algorithms usually maintain probability models explicitly, which are then plugged into an AC module (Rissannen, 1976; Pasco, 1976; Rissanen & Langdon, 1979) for efficient compression. The most well-known classes of algorithms in this category is the context-tree weighting algorithm (Willems et al., 1995; Begleiter et al., 2004; Kontoyiannis, 2023) and prediction by partial matching (Cleary & Witten, 1984). The former enjoys a strong theoretical guarantee, particularly on binary sources (Willems et al., 1995), but has some difficulty in its practical implementation (Willems, 1998; Willems et al., 1996; Sadakane et al., 2000; Begleiter et al., 2004), particularly for large alphabet sizes and sequential data. The latter is based more on heuristics, and has been improved and extended in various ways (Cleary & Teahan, 1997; Moffat, 1990; Shkarin, 2002). Methods based on probabilistic modeling are usually more resource-extensive, though they have gained more popularity recently due to the increased availability of computing resources. The evaluation given in (Begleiter et al., 2004) suggests that CTW and PPM are the two most powerful compression algorithms in practice. There are other compression

algorithms such as those based on the Burrows-Wheeler transformation (Burrows, 1994) which does not explicitly maintain a probabilistic model, but are also not dictionary-based.

# B  OTHER PRELIMINARIES

## B.1  TRANSFORMER ARCHITECTURE

The transformer architecture considered in this work is illustrated in Fig. 10.

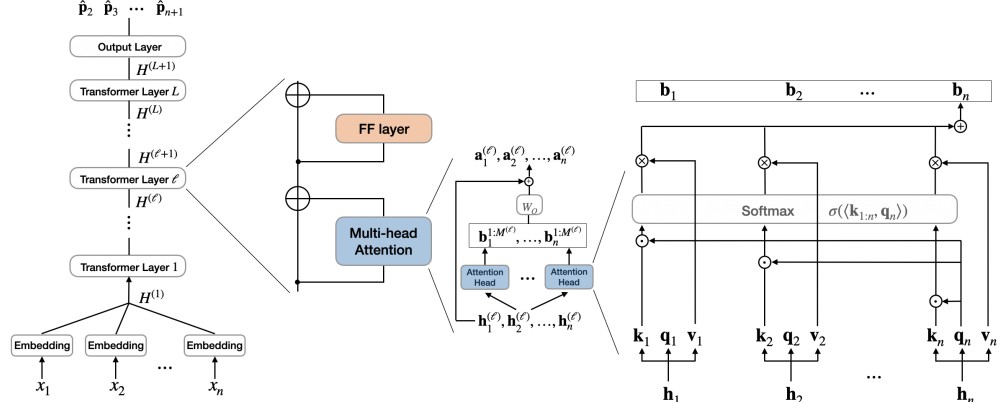

Figure 10: Transformer model

## B.2  THE PPM ALGORITHM

The PPM algorithm (with finite memory of parameter $D_{\text{PPM}}$) blends several CTs by utilizing an escape symbol (Esc), and adaptively refines the CT model using the observed samples. The key idea is that the estimated probability distribution for an emitted symbol is only used when there were past observations of this string. For other cases, the escape symbol is encoded, indicating a shorter suffix needs to be used.

Table 2: PPM counts after observing string $(a, b, c, a, b, b, c)$

| order $k = 2$ | | | order $k = 1$ | | | order $k = 0$ | | | order $k = -1$ | | |
|---|---|---|---|---|---|---|---|---|---|---|---|
| prediction | $c$ | $p$ | prediction | $c$ | $p$ | prediction | $c$ | $p$ | prediction | $c$ | $p$ |
| $(a,b) \rightarrow b$ | 1 | $\frac{1}{3}$ | $a \rightarrow b$ | 2 | $\frac{2}{3}$ | $\rightarrow a$ | 2 | $\frac{1}{4}$ | | | $\frac{1}{|\mathcal{A}|}$ |
| $\rightarrow c$ | 1 | $\frac{1}{3}$ | $\rightarrow$ Esc | 1 | $\frac{1}{3}$ | $\rightarrow b$ | 3 | $\frac{3}{8}$ | | | |
| $\rightarrow$ Esc | 1 | $\frac{1}{3}$ | $b \rightarrow b$ | 1 | $\frac{1}{4}$ | $\rightarrow c$ | 2 | $\frac{1}{4}$ | | | |
| $(b,b) \rightarrow c$ | 1 | $\frac{1}{2}$ | $\rightarrow c$ | 2 | $\frac{1}{2}$ | $\rightarrow$ Esc | 1 | $\frac{1}{8}$ | | | |
| $\rightarrow$ Esc | 1 | $\frac{1}{2}$ | $\rightarrow$ Esc | 1 | $\frac{1}{4}$ | | | | | | |
| $(b,c) \rightarrow a$ | 1 | $\frac{1}{2}$ | $c \rightarrow a$ | 1 | $\frac{1}{2}$ | | | | | | |
| $\rightarrow$ Esc | 1 | $\frac{1}{2}$ | $\rightarrow$ Esc | 1 | $\frac{1}{2}$ | | | | | | |
| $(c,a) \rightarrow c$ | 1 | $\frac{1}{2}$ | | | | | | | | | |
| $\rightarrow$ Esc | 1 | $\frac{1}{2}$ | | | | | | | | | |

We illustrate this context tree blending approach by the example shown in Table. 2, where $\mathcal{A} = \{a, b, c\}$, and the memory length $D_{\text{PPM}} = 2$. The escape pattern is assigned a count one (method-A in (Moffat, 1990)). Suppose the next symbol to emit is $a$, then the probability prediction is $\frac{1}{2}$ from the $k = 2$ column; if on the other hand, the next symbol to emit is $b$, then the escape symbol is first encoded with probability $\frac{1}{2}$ since there is no string of $(b, c, b)$ in the history, and then we check the column $k = 1$, and see that another escape symbol will be encoded since there is also no $(c, b)$, and finally $b$ will be encoded at $k = 0$, and the eventual effective probability for $b$ is $\frac{1}{2} \cdot \frac{1}{2} \cdot \frac{3}{8}$. Various refinements of the probability estimation can be adopted to further improve the performance, e.g., the exclusion rule, and other methods to initialize the probability for Esc; see e.g. (Moffat, 1990; Cleary & Teahan, 1997; Begleiter et al., 2004). As the number of observed samples accumulated, all patterns of $(D_{\text{PPM}} + 1)$-grams will be observed at least once and the probability prediction will solely based on the column of maximum order $k = D_{\text{PPM}}$.

## C    PRETRAINING DETAILS

We choose the alphabet size to be $|\mathcal{A}| = 3$ in the experiments. For training, we randomly generate $K = 20000$ CTs of various depths (maximum order $D \leq 5$), and then for each CT leaf, we generate a probability distribution. Two different ways of generating these probability distributions are taken: the first approach is use the Dirichlet distribution to sample such distributions, and the second approach is to randomly select some of the elements in the alphabet to have probability zero, and the others with i.i.d. random values before normalization. Different values of the Dirichlet parameter are tested but only the results do not appear to be sensitive to the choice. For each CT, a source sequence of certain length (e.g., $N_k = 5120$) is produced. The context window $N$ can vary, but in most cases, we set it at 512 (except when $D = 5$, we set it to be 1536 to allow sufficient data collection in context). Each source sequence is segmented into $\lfloor N_k/N \rfloor$ training sequence.

During testing, we randomly generate multiple (2048 in our experiments) new CTs of varying depths using the same procedure, and for each CT, a sequence of length $N_k = 5120$ are generated, and then again segmented into a length of the context window for testing.

The transformer model is implemented using Pytorch, and trained using the AdamW optimizer with the default parameters. A100/T100 GPUs are used for training. Training a model requires roughly 4 to 6 hours. Batch size is set at 512, and the maximum epoch is set at 100 with early termination allowed after 15 epochs of no improvement. Testing was performed on a local workstation with a GeForce GTX 1660 Ti GPU card.

## D    PROOFS OF THE THEOREMS FOR CT SOURCES

### D.1    A NEW FORMULA FOR BAYESIAN NEXT TOKEN PREDICTION

We aim to predict the next token $x_{n+1}$ based on the observations $x_{1-D}^n = (x_{1-D}, \ldots, x_n)$ via a transformer-friendly formula. Note that $x_{1-D}^0$ is a place holder or dummy initialization sequence, which does not contain any information of the CT $(T, \{p_s\})$; or alternatively, we can view $P(T, \{p_s\}|x_{1-D}^0) = \pi_{\text{CTW}}(T, \{p_s\})$ parameterized by $\lambda, \boldsymbol{\alpha}$.

**Theorem 6** (Restate Theorem 2)**.** *The predicted probability can be computed as*

$$P_{\pi_{CTW}}(x_{n+1}|x_{1-D}^n) = \sum_{l=0,\ldots,D} \omega_{n,l} \cdot \mathbf{p}_{n,s_{n,l}}(x_{n+1}), \tag{13}$$

*where* $\mathbf{p}_{n,s_{n,l}}(a) = \frac{\boldsymbol{\alpha}(a) + \mathbf{n}_{n,s_{n,l}}(a)}{\sum_q (\boldsymbol{\alpha}(q) + \mathbf{n}_{n,s_{n,l}}(q))}$; *and* $\omega_{n,\cdot} \in \Delta_{D+1}$ *with* $\ln(\omega_{n,l}) - \ln(\omega_{n,l-1}) = \ln(1-\lambda) -$
$\mathbb{I}_{l=D} \ln(\lambda) + \ell_{n,s_{n,l}}^e - \ell_{n,s_{n,l-1}}^e + \sum_{q \in \mathcal{A}} \ell_{n,qs_{n,l-1}}^w - \ell_{n,s_{n,l}}^w$, *where* $\ell_{n,s}^e = \ln(p_{n,s}^e)$, $\ell_{n,s}^w = \ln(p_{n,s}^w)$.

**Discussion.** Note that $p_{n,s}^e, p_{n,s}^w$ can be efficiently calculated by the CTW procedure, and compared to calculate $\frac{P_{\pi_{\text{CTW}}}(x_1^{n+1}|x_{1-D}^0)}{P_{\pi_{\text{CTW}}}(x_1^n|x_{1-D}^0)}$ for each $x_{n+1}$ the extra computation besides the CTW procedure is $A$ times larger than that by Eq (7). As illustrated in Fig. 8, the weighted average formula in Eq (7) gives a natural interpretation for the Bayesian optimal next token predicted probability. Each suffix along the root the leaf path $s_{n,0} - s_{n,1} - \cdots - s_{n,D}$ can potentially be the true suffix, i.e., $s_{n,l} \in \mathcal{L}(T)$, and $\mathbf{p}_{n,s_{n,l}}$ is in fact the Bayesian optimal next token prediction given $s_{n,l} \in \mathcal{L}(T)$.

The blending weights $\omega_{n,l}$'s are based on stopping probability $\lambda$, the information in the potential suffix path such as $p_{s_{n,s_{n,l}}}^e$ as well as the information from their siblings $p_{n,qs_{n,l-1}}^w$. We can interpret $p_{n,s}^e$ as the evidence (unnormalized likelihood) that $s \in \mathcal{L}(T)$, and $p_{n,s}^w$ as the evidence that $s \in T$, i.e., the underlying tree covers node $s$. Theorem 6 indicates that more weights are assigned to $s_{n,l}$ than $s_{n,l-1}$, i.e., $\omega_{n,l} > \omega_{n,l-1}$, if $\lambda$ is smaller (i.e., node $s_{n,l-1}$ is more likely to branch and thus less likely to be a leaf node), $p_{n,s_{n,l}}^e - p_{n,s_{n,l-1}}^e$ is larger (i.e., $s_{n,l}$ has more evidence than $s_{n,l-1}$) and $\sum_{q \in \mathcal{A}} \ell_{n,qs_{n,l-1}}^w - \ell_{n,s_{n,l}}^w$ is larger (i.e., $s_{n,l}$'s siblings have more evidence to explain the data and thus $s_{n,l-1}$ is less likely to be a leaf node). The indicator function $\mathbb{I}_{l=D}$ is due to the maximum depth constraint on the branching process. Nodes at level $l = D$ automatically stop the branching, i.e., the branching-stopping probability is 1 for such nodes.

*Proof of Theorem 6.* Recall $s_{i,l} = (x_{i-l+1}, \ldots, x_i)$ is the suffix at position $i$ of length $l$. We omit $D$ by writing $\mathcal{T} = \mathcal{T}(D)$ when $D$ is clear from the context. Define partition $\{\mathcal{T}_{s_{n,l}}\}_{0 \le l \le D}$, that $\mathcal{T}_s = \{T \in \mathcal{T} : s \in \mathcal{L}(T)\}$ is the set of trees containing leaf $s$. The predicted probability can then be computed as

$$P_{\pi_{\mathrm{CTW}}}(x_{n+1}|x_{1-D}^n) = \sum_{T \in \mathcal{T}} \int p(x_{n+1}|T, \{p_s\}, x_{1-D}^n) \pi_{\mathrm{CTW}}(T, \{p_s\}|x_{1-D}^n) \Big( \prod_{s \in \mathcal{L}(T)} \mathrm{d}p_s \Big)$$

$$= \sum_{l=0,\ldots,D} \sum_{T \in \mathcal{T}_{s_{n,l}}} \int p_{s_{n,l}}(x_{n+1}) \pi_{\mathrm{CTW}}(T, \{p_s\}|x_{1-D}^n) \Big( \prod_{s \in \mathcal{L}(T)} \mathrm{d}p_s \Big)$$

$$= \sum_{l=0,\ldots,D} \sum_{T \in \mathcal{T}_{s_{n,l}}} \int p_{s_{n,l}}(x_{n+1}) \pi_D(T|x_{1-D}^n) \pi_p(p_{s_l}|T, x_{1-D}^n) \mathrm{d}p_{s_l}$$

$$= \sum_{l=0,\ldots,D} \sum_{T \in \mathcal{T}_{s_{n,l}}} \pi_D(T|x_{1-D}^n) \int p_{s_{n,l}}(x_{n+1}) \pi_p(p_{s_l}|T, x_{1-D}^n) \mathrm{d}p_{s_l}$$

$$= \sum_{l=0,\ldots,D} \Big( \sum_{T \in \mathcal{T}_{s_{n,l}}} \pi_D(T|x_{1-D}^n) \Big) \Big( \int p_{s_{n,l}}(x_{n+1}) \pi_p(p_{s_l}|T, x_{1-D}^n) \mathrm{d}p_{s_l} \Big)$$

$$= \sum_{l=0,\ldots,D} \omega_{n,l} \cdot \mathbf{p}_{n,s_{n,l}}(x_{n+1}), \tag{14}$$

where the last equality is by the definition that

$$\omega_{n,l} = \sum_{T \in \mathcal{T}_{s_{n,l}}} \pi_D(T|x_{1-D}^n), \tag{15}$$

and the optimal prediction probability given suffix $s_{n,l}$ is

$$\mathbf{p}_{n,s_{n,l}}(a) = \frac{\boldsymbol{\alpha}(a) + \mathbf{n}_{n,s_{n,l}}(a)}{\sum_{q \in \mathcal{A}}(\boldsymbol{\alpha}(q) + \mathbf{n}_{n,s_{n,l}}(q))}, \tag{16}$$

since for any $T \in \mathcal{T}_{s_l}$, the posterior of $p_s$ follows Dirichlet distribution

$$\pi_p(p_{s_l}|T, x_{1-D}^n) = \mathrm{Dir}(p_{s_l}; \boldsymbol{\alpha} + \mathbf{n}_{n,s_{n,l}}), \tag{17}$$

with posterior mean $\mathbb{E}[p_{s_l}|T, x_{1-D}^n] \in \Delta_{\mathcal{A}}$ and proportional to $\boldsymbol{\alpha} + \mathbf{n}_{n,s_{n,l}}$.

Since the length of data $n$ is fixed and clear from the context, let $\underline{x} = x_{1-D}^n$ be the sequence, and we omit $n$ in the subscript of $p_{n,s}^e, p_{n,s}^w$ and $s_{n,l}$ for simplicity.

For any model $T \in \mathcal{T}(D)$, the posterior probability $\pi(T|\underline{x})$ is given by:

$$\pi_D(T|\underline{x}) = \frac{\pi_D(T) P_\pi(\underline{x}|T)}{P_\pi(\underline{x})} = \frac{\pi_D(T) \prod_{s \in \mathcal{L}(T)} p_s^e}{p_{()}^w}, \tag{18}$$

where the denominator $P_\pi^*(\underline{x}) = p_{()}^w$ is the prior predictive likelihood computed by CTW given in Theorem 1, and the numerator is by $P_\pi(\underline{x}|T) = \prod_{s \in \mathcal{L}(T)} p_s^e$ in (Kontoyiannis et al., 2022, Lemma 2.2). Since $\omega_l = \sum_{T \in \mathcal{T}_{s_l}} \pi(T|\underline{x})$ by definition, we have for any $l = 1, 2, \ldots, d$,

$$\frac{\omega_l}{\omega_{l-1}} = \frac{\sum_{T' \in \mathcal{T}_{s_l}} \pi_d(T'|x)}{\sum_{T \in \mathcal{T}_{s_{l-1}}} \pi_d(T|x)} = \frac{\sum_{T' \in \mathcal{T}_{s_l}} \pi_d(T') \prod_{s \in \mathcal{L}(T')} p_s^e}{\sum_{T \in \mathcal{T}_{s_{l-1}}} \pi_d(T) \prod_{s \in \mathcal{L}(T)} p_s^e}. \tag{19}$$

Note that tree in $\mathcal{T}_{s_l}$ and trees in $\mathcal{T}_{s_{l-1}}$ share similarities. For any $T \in \mathcal{T}_{s_{l-1}}$, let $\mathcal{T}_{s_l;T} = \{T' \in \mathcal{T}_{s_l} : \mathcal{L}(T) \subset \mathcal{L}(T') \cup \{s_{l-1}\}\}$ be the set of trees that differs from $T$ only at subtree $\mathrm{sub}(T'; s_l) := \{\text{subtree of } T' \text{ with root at } s\}$.

Take any $l = 1, 2, \ldots, D-1$. For any $T \in \mathcal{T}_{s_{l-1}}$ and $T' \in \mathcal{T}_{s_l;T}$. Based on the definition of $\pi_D = (1-\lambda)^{(|\mathcal{L}(T)|-1)/(A-1)} \lambda^{|\mathcal{L}(T)|-|\mathcal{L}_D(T)|}$, it is not hard to verify that

$$\frac{\pi_D(T')}{\pi_D(T)} = \frac{\pi_{D-l+1}(\mathrm{sub}(T'; s_{l-1}))}{\pi_{D-l+1}(\mathrm{sub}(T; s_{l-1}))}$$

$$= \frac{(1-\lambda)\pi_{D-l}(\text{sub}(T';s_l)) \prod_{s'_l \in \text{sib}(s_l)} \pi_{D-l}(\text{sub}(T';s'_l))}{\lambda}$$

$$= (1-\lambda) \prod_{s'_l \in \text{sib}(s_l)} \pi_{D-l}(\text{sub}(T';s'_l)),$$

where $\text{sib}(s_{l+1}) = \{qs_l : q \in \mathcal{A} \text{ and } qs_l \neq s_{l+1}\}$ is set of siblings of $s_{l+1}$. We can interpret the ratio as follows. $T'$ and $T$ only differs at the $\text{sub}(T';s_{l-1})$ and $\text{sub}(T;s_{l-1})$. Since $T'$ branch at node $s_{l-1}$, we thus have the numerator in the second equation, where $(1-\lambda)$ corresponds to the branching and then compute for the subtrees. Note that $T$ stops branching at $s_{l-1}$ and $T'$ stops branching at $s_l$, then $\pi_{D-l+1}(\text{sub}(T;s_{l-1})) = \pi_{D-l}(\text{sub}(T';s_l)) = \lambda$ equals to the stopping probability.

Given any suffix $s$ with $|s| \leq D$, it has been shown in (Kontoyiannis et al., 2022, Proof of Theorem 3.1) that for any $l \leq D$,

$$p_s^w = \sum_{U \in \mathcal{T}(D-l)} \pi_{D-l}(U) \prod_{u \in \mathcal{L}(U)} p_{us}^e, \tag{20}$$

where $\mathcal{T}(D-l)$ is the set of trees with maximum depth $D-l$ and $\pi_{D-l}$ is the prior for bounded branching process with maximum depth $D-l$. We thus have

$$\frac{\sum_{T' \in \mathcal{T}_{s_l;T}} \pi_D(T') \prod_{s \in \mathcal{L}(T')} p_s^e}{\pi_D(T) \prod_{s \in \mathcal{L}(T)} p_s^e} = \frac{\sum_{T' \in \mathcal{T}_{s_l;T}} \pi_D(T') \prod_{s \in \mathcal{L}(T')} p_s^e}{\pi_D(T) \prod_{s \in \mathcal{L}(T)} p_s^e} \tag{21}$$

$$= \sum_{T' \in \mathcal{T}_{s_l;T}} \frac{\pi_D(T')}{\pi_D(T)} \frac{\prod_{s \in \mathcal{L}(T') \backslash \mathcal{L}(T)} p_s^e}{p_{s_{l-1}}^e} \tag{22}$$

$$= \sum_{T' \in \mathcal{T}_{s_l;T}} \left( (1-\lambda) \prod_{s'_l \in \text{sib}(s_l)} \pi_{D-l}(\text{sub}(T';s'_l)) \right) \left( \frac{p_{s_l}^e \prod_{s'_l \in \text{sib}(T;s_l)} \prod_{s \in \mathcal{L}(\text{sub}(T';s'_l))} p_s^e}{p_{s_{l-1}}^e} \right) \tag{23}$$

$$= (1-\lambda) \frac{p_{s_l}^e}{p_{s_{l-1}}^e} \sum_{T' \in \mathcal{T}_{s_l;T}} \left( \prod_{s'_l \in \text{sib}(s_l)} \pi_{D-l}(\text{sub}(T';s'_l)) \right) \left( \prod_{s'_l \in \text{sib}(T;s_l)} \prod_{s \in \mathcal{L}(\text{sub}(T';s'_l))} p_s^e \right) \tag{24}$$

$$= (1-\lambda) \frac{p_{s_l}^e}{p_{s_{l-1}}^e} \sum_{T' \in \mathcal{T}_{s_l;T}} \left( \prod_{s'_l \in \text{sib}(s_l)} \pi_{D-l}(\text{sub}(T';s'_l)) \prod_{s \in \mathcal{L}(\text{sub}(T';s'_l))} p_s^e \right) \tag{25}$$

$$= (1-\lambda) \frac{p_{s_l}^e}{p_{s_{l-1}}^e} \prod_{s'_l \in \text{sib}(s_l)} \left( \sum_{U \in \mathcal{T}(D-l)} \pi_{D-l}(U) \prod_{u \in \mathcal{L}(U)} p_{us'_l}^e \right) \tag{26}$$

$$= \frac{(1-\lambda)p_{s_l}^e \prod_{a \neq s_l \backslash s_{l-1}} p_{as_{l-1}}^w}{p_{s_{l-1}}^e}. \tag{27}$$

Similarly, for any $T \in \mathcal{T}_{s_{D-1}}$ and $T' \in \mathcal{T}_{s_D;T}$, $\frac{\pi_D(T')}{\pi_D(T)} = \frac{1-\lambda}{\lambda}$, we can derive

$$\frac{\omega_D}{\omega_{D-1}} = \frac{(1-\lambda)p_{s_d}^e \prod_{a \neq s_d \backslash s_i} p_{as_{D-1}}^w}{\lambda p_{s_{D-1}}^e}, \tag{28}$$

in the same manner. The proof can then be concluded by taking the logarithm on both hands. $\square$

## D.2 Construction of Transformer for CTW

To make the presentation clear, in the following we separate the layers by their functionality and present them separately. Recall that

$$\mathbf{a}_i^{(\ell)} = \text{MHA}\left(\mathbf{h_i}, \mathbf{H}; \{W_{O,m}^{(\ell)}, W_{Q,m}^{(\ell)}, W_{K,m}^{(\ell)}, W_{V,m}^{(\ell)}\}_{m=1}^{M^{(\ell)}}\right) \triangleq W_O^{(\ell)} \left[\mathbf{b}_{1,i}^{(\ell)}; \mathbf{b}_{2,i}^{(\ell)}; \ldots; \mathbf{b}_{M^{(\ell)},i}^{(\ell)}\right],$$

where $\{W_{Q,m}^{(\ell)}, W_{K,m}^{(\ell)}, W_{V,m}^{(\ell)}\}_{m=1}^{M^{(\ell)}}$ are the $E^{(\ell)} \times E$ query matrices, key matrices, and value matrices and $W_O^{(\ell)}$ is the $E \times M^{(\ell)} E^{(\ell)}$ output mapping matrix. For simplicity of presentation, we take

$E^\ell = E$ and $W_O^\ell = [\mathbf{I}; \mathbf{I}; \ldots; \mathbf{I}]$. It is not hard to see the following constructions can be applied to much smaller $E^{(\ell)}$ while taking $W_O$ as a permutation matrix.

We have omitted the dimensionlity of several zero matrices when they are obvious from the context. The first and second layer constructions are illustrated in Fig. 11.

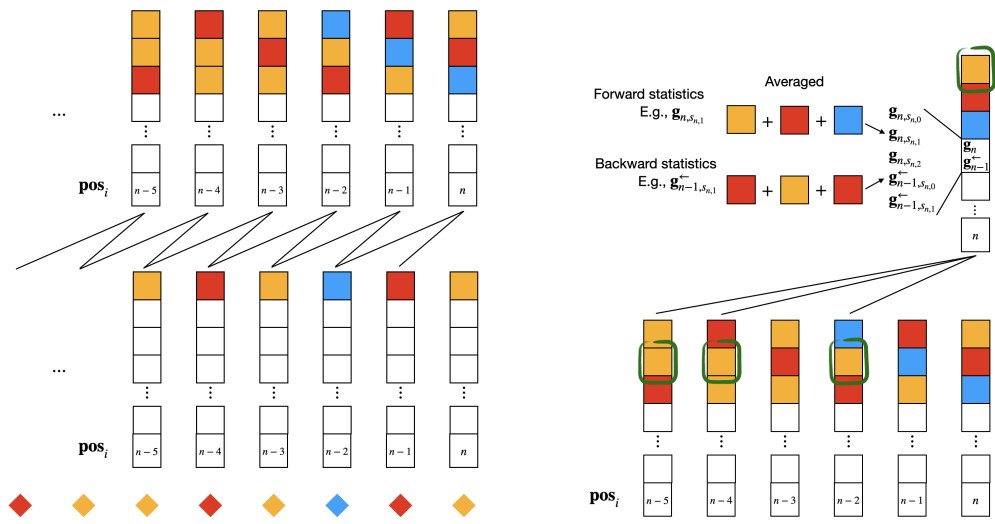

Figure 11: Transformer construction for $D = 2$. The left figure illustrates the first layer – finite-memory context-extension layer, which append the previous $D$ tokens. The right figure demonstrate the MHA of the second layer – statistics collection layer, which extracts forward and backward statistics based on the matched suffix indicated by a green square.

### D.2.1  FINITE-MEMORY CONTEXT-EXTENSION LAYER

We begin with the first layer, which is referred to as a finite-memory context-extension layer.

**Theorem 7** (Restate Theorem 3). *There is an $M$-headed transformer layer that can perform finite-memory context-extension, defined by the following output, with the initial one-hot embedded input $\mathbf{H}^{(1)}$:*

$$\mathbf{h}_i^{(2)} = (\mathbf{s}_{i,M+1}; \mathbf{0}; \mathbf{pos}_i) = (\mathbf{x}_i; \mathbf{x}_{i-1}; \ldots; \mathbf{x}_{i-M}; \mathbf{0}; \mathbf{pos}_i), \tag{29}$$

*where $\mathbf{s}_{i,M+1} = (\mathbf{x}_i; \ldots; \mathbf{x}_{i-M})$ is the vector version of the $M$-length suffix $s_{i,M+1} = x_{i-M}^i$.*

*Proof of Theorem 3.* The input of the of the first layer is a initial one-hot embedded input with positional embedding $\mathbf{H}^{(1)}$, where its $n$-th column is

$$\mathbf{h}_i^{(1)} = (\mathbf{x}_i; \mathbf{0}; \mathbf{pos}_i) \in \mathbb{R}^E, \tag{30}$$

where positional encoding

$$\mathbf{pos}_i = (1; \cos(i\pi/N); \sin(i\pi/N)), \tag{31}$$

where $N$ is the maximum context window size.

The multi-head attention in the first layer is consisted of $M^{(1)} = D$ heads parameterized by $(W_{Q,m}^{(1)}, W_{K,m}^{(1)}, W_{V,m}^{(1)})_{m=1,2,\ldots,M^{(1)}}$. Specifically, for $m = 1, 2, \ldots, M^{(1)}$,

$$W_{Q,m}^{(1)} = \begin{pmatrix} \mathbf{0} & \mathrm{Rot}(m) \\ \mathbf{0} & \mathbf{0} \end{pmatrix}, \quad W_{K,m}^{(1)} = \begin{pmatrix} \mathbf{0} & c\mathbf{I}^{2\times2} \\ \mathbf{0} & \mathbf{0} \end{pmatrix}, \quad W_{V,m}^{(1)} = \begin{pmatrix} \mathbf{0}^{mA\times A} & \mathbf{0} \\ \mathbf{I}^{A\times A} & \mathbf{0} \\ \mathbf{0} & \mathbf{0} \end{pmatrix}, \tag{32}$$

where $\text{Rot}(m) = \begin{pmatrix} \cos(m\pi/N) & \sin(m\pi/N) \\ -\sin(m\pi/N) & \cos(m\pi/N) \end{pmatrix}$ is a rotation matrix that rotates clockwise by an angle of $m\pi/C$, and $c \in \mathbb{R}_+$ is a temperature factor. The query, key, and value after the mapping are

$$W_{Q,m}^{(1)} \mathbf{h}_n^{(1)} = \begin{pmatrix} \mathbf{pos}_{n-m} \\ \mathbf{0} \end{pmatrix}, \quad W_{K,m}^{(1)} \mathbf{h}_i^{(1)} = c \begin{pmatrix} \mathbf{pos}_i \\ \mathbf{0} \end{pmatrix}, \quad W_{V,m}^{(1)} \mathbf{h}_i^{(1)} = \begin{pmatrix} \mathbf{0}^{mA \times 1} \\ \mathbf{x}_i \\ \mathbf{0} \end{pmatrix}. \tag{33}$$

Take $c = \infty$ or sufficiently large. It is seen that the $m$-th head essentially copies the $m$-th earlier symbol to stack at the $(m+1)$-th position below the original symbol $\mathbf{x}_i$. Together with the residual link, the attention layer gives exactly the $\mathbf{h}_i^{(2)}$ shown in (34) while the FF layer in this layer can be set as all zero.

$$\mathbf{h}_i^{(2)} = (\mathbf{x}_i; \mathbf{x}_{i-1}; \mathbf{x}_{i-2}; \mathbf{x}_{i-M^{(1)}}; \mathbf{0}; \mathbf{pos}_i) = (\mathbf{s}_{i,M^{(1)}+1}; \mathbf{0}; \mathbf{pos}_i), \tag{34}$$

where $\mathbf{s}_{i,l} = (\mathbf{x}_i; \mathbf{x}_{i-1}; \cdots; \mathbf{x}_{i-l+1})$ is the one-hot embedded version of suffix $s_{i,l} = (x_{i-l+1}, \ldots, x_{i-1}, x_i)$.  □

### D.2.2 STATISTICS COLLECTION LAYER

**Theorem 8** (Restate Theorem 4). *There is an $M'$-head attention layer, where $M' \leq M+1$, that can perform statistics collection, defined by the following output, with $\mathbf{H}^{(2)}$ in (8) as its input:*

$$\mathbf{a}_i^{(2)} = (\mathbf{s}_{i,M+1}; \mathbf{g}_{i,M'}; \mathbf{g}_{i-1,M'}^{\leftarrow}; \mathbf{0}; \mathbf{pos}_i), \tag{35}$$

*where $\mathbf{g}_{i,M'} := (\mathbf{g}_{i,s_{i,0}}; \ldots; \mathbf{g}_{i,s_{i,M'-1}})$ and $\mathbf{g}_{i-1,M'}^{\leftarrow} = (\mathbf{g}_{i-1,s_{i,0}}^{\leftarrow}; \ldots; \mathbf{g}_{i-1,s_{i,M'-1}}^{\leftarrow})$.*

*Proof of Theorem 4.* To make the proof self-contained, we first recall some key notations. The second layer is referred to as the statistics collection layer, which uses a sequence of vectors $\mathbf{h}_i^{(2)}$, $i = 1, 2, \ldots, N$, defined in (8) as its input, restated as follows.

$$\mathbf{h}_i^{(2)} = (\mathbf{s}_{i,M+1}; \mathbf{0}; \mathbf{pos}_i), \tag{36}$$

where $\mathbf{s}_{i,M+1} = (\mathbf{x}_i; \ldots; \mathbf{x}_{i-M})$. To rigorously specify the function of this layer, recall the definition of the $k$-gram statistics vector $\mathbf{g}_{i,s}$, which in plain words, is the empirical probability distribution of the next token associated with the suffix $s$ for a sequence $x_1^i$. Mathematically, for a suffix $s$ whose length is $k-1$ and the current position $i$,

$$\mathbf{g}_{i,s}(a) = \frac{\mathbf{n}_{i,s}(a)}{\sum_{q \in \mathcal{A}} \mathbf{n}_{i,s}(q)} \quad \forall a \in \mathcal{A}, \tag{37}$$

where $\mathbf{n}_{i,s}$ is the counting vector defined in (5).

The $k$-gram backward statistics vector $\mathbf{g}_{i-1,s}^{\leftarrow}$ is defined similarly, which is the empirical probability distribution of the previous token associated with the suffix $s$ for data $x_1^{i-1}$, and mathematically

$$\mathbf{g}_{i-1,s}^{\leftarrow}(a) = \frac{\sum_{q \in \mathcal{A}} \mathbf{n}_{i,as}(q)}{\sum_{q \in \mathcal{A}} \mathbf{n}_{i,s}(q)} \quad \forall a \in \mathcal{A}, \tag{38}$$

where $\sum_{q \in \mathcal{A}} \mathbf{n}_{i,s}(q)$ is the number of appears of the sub-string $s$ in the sequence $x_1^{i-1}$.

The multi-head attention in the second layer is consisted of $M^{(2)} = M' \leq M^{(1)} + 1 = M + 1$ heads parameterized by $(W_{Q,m}^{(2)}, W_{K,m}^{(2)}, W_{V,m}^{(2)})_{m=0,1,2,\ldots,M^{(2)}-1}$. Specifically, for $m = 1, 2, \ldots, M^{(2)} - 1$,

$$W_{Q,m}^{(2)} = \begin{pmatrix} \mathbf{I}^{(m-1)A \times (m-1)A} & \mathbf{0} \\ \mathbf{0} & \mathbf{0} \end{pmatrix}, \quad W_{K,m}^{(2)} = \begin{pmatrix} \mathbf{0}^{(m-1)A \times A} & c\mathbf{I}^{(m-1)A \times (m-1)A} & \mathbf{0} \\ \mathbf{0} & \mathbf{0} & \mathbf{0} \end{pmatrix}, \tag{39}$$

$$W_{V,m}^{(2)} = \begin{pmatrix} \mathbf{0}^{(M^{(1)}+m)A \times A} & \mathbf{0} \\ \mathbf{I}^{A \times A} & \mathbf{0} \\ \mathbf{0}^{(M^{(2)}-1)A \times A} & \mathbf{0} \\ \mathbf{0}^{A \times A} & [\mathbf{0}^{A \times (m-1)A}, \mathbf{I}^{A \times A}, \mathbf{0}] \\ \mathbf{0} & \mathbf{0} \end{pmatrix}. \tag{40}$$

The corresponding query, key, and value vectors after the mapping are

$$W_{Q,m}^{(2)}\mathbf{h}_n^{(2)} = \begin{pmatrix}\mathbf{s}_{n,m-1}\\\mathbf{0}\end{pmatrix}, \quad W_{K,m}^{(2)}\mathbf{h}_i^{(2)} = c\begin{pmatrix}\mathbf{s}_{i-1,m-1}\\\mathbf{0}\end{pmatrix}, \quad W_{V,m}^{(2)}\mathbf{h}_i^{(2)} = \begin{pmatrix}\mathbf{0}^{(M^{(1)}+m)A\times 1}\\\mathbf{x}_i\\\mathbf{0}^{(M^{(2)}-1)A\times 1}\\\mathbf{x}_{i-m}\\\mathbf{0}\end{pmatrix}.$$

For $m = M^{(2)}$, $W_{Q,m}^{(2)}, W_{K,m}^{(2)}$ are of the same structure, while $W_{V,m}^{(2)}$ does not contains that $\mathbf{I}^{A\times A}$ in that $[\mathbf{0}^{A\times(m-1)A}, \mathbf{I}^{A\times A}, \mathbf{0}]$ block, and thus $W_{V,m}^{(1)}\mathbf{h}_i^{(1)}$ does not have $\mathbf{x}_{i-m}$.

It is not hard to see that taking $c \to \infty$ gives

$$(\mathbf{s}_{i,M^{(1)}+1}; \mathbf{g}_{i,M^{(2)}-1}; \mathbf{g}_{i-1,M^{(2)}-1}^{\leftarrow}; \mathbf{0}; \mathbf{pos}_i) = [\text{MHA}(\mathbf{H}^{(2)}) + \mathbf{H}^{(2)}]_i, \tag{41}$$

where

$$\mathbf{g}_{i,M'} = (\mathbf{g}_{i,s_{i,0}}; \ldots; \mathbf{g}_{i,s_{i,M'-1}})$$
$$\mathbf{g}_{i-1,M'}^{\leftarrow} = (\mathbf{g}_{i-1,s_{i,0}}^{\leftarrow}; \ldots; \mathbf{g}_{i-1,s_{i,M'-1}}^{\leftarrow}).$$

$\square$

Note that the counting vector can be obtained via

$$\mathbf{n}_{i,s_{i,l}}(a) = \frac{\mathbf{n}_{i,s_{i,l}}(a)}{\sum_{q\in\mathcal{A}}\mathbf{n}_{i,s_{i,l}}(q)}\frac{\sum_{q\in\mathcal{A}}\mathbf{n}_{i,s_{i,l}}(q)}{\sum_{q\in\mathcal{A}}\mathbf{n}_{i,s_{i,l-1}}(q)}\cdots\frac{\sum_{q\in\mathcal{A}}\mathbf{n}_{i,s_{i,1}}(q)}{\sum_{q\in\mathcal{A}}\mathbf{n}_{i,s_{i,0}}(q)}\left(\sum_{q\in\mathcal{A}}\mathbf{n}_{i,s_{i,0}}(q)\right) \tag{42}$$

$$= \mathbf{g}_{i,s_{i,l}}(a)\left(\prod_{j=0}^{l-1}\mathbf{g}_{i-1,s_{i,j}}^{\leftarrow}(x_{i-j})\right)\cdot i, \tag{43}$$

by the information contained in vector $(\mathbf{s}_{i,M^{(1)}+1}; \mathbf{g}_{i,M^{(2)}-1}; \mathbf{g}_{i-1,M^{(2)}-1}^{\leftarrow}; \mathbf{0}; \mathbf{pos}_i)$.

Since $p_{i,s_{i,l}}^e$ and $\mathbf{p}_{i,s_{i,l}}$ in (16) are functions of $\mathbf{n}_{i,s_{i,l}}$, we can then obtain (approximate) the following output by a sufficiently wide FF layer that

$$\mathbf{h}_i^3 = (\mathbf{s}_{i,M^{(1)}+1}; \mathbf{p}_{i,D}; \mathbf{l}_{i,D}^e; \ln(p_{i,s_{i,D}}^w); \mathbf{0}; \mathbf{pos}_i), \tag{44}$$

where $\mathbf{l}_{i,D}^e$ contains the logarithm of $p^e$ along the path from root () to $(x_{i-d+1}, \ldots, x_i)$, and $\mathbf{p}_{i,D}$ stacks the optimal prediction given suffices $s_{i,0}, \ldots, s_{i,D}$, i.e.,

$$\mathbf{l}_{i,D}^e = (\ell_{i,s_{i,0}}^e; \ell_{i,s_{i,1}}^e; \ldots; \ell_{i,s_{i,D}}^e) = (\ln(p_{i,s_{i,0}}^e); \ln(p_{i,s_{i,1}}^e); \ldots; \ln(p_{i,s_{i,D}}^e)), \tag{45}$$

$$\mathbf{p}_{i,D} = (\mathbf{p}_{i,s_{i,0}}; \mathbf{p}_{i,s_{i,1}}; \ldots; \mathbf{p}_{i,s_{i,D}}), \tag{46}$$

and $\ln(p_{i,s_{i,D}}^w) = \ln(p_{i,s_{i,D}}^e)$ with suffix $|s_{i,D}| = D$. These quantities can be extracted, since they are functions of the statistics collected from $\mathbf{a}_i^{(2)}$.

This functional layer essentially collects $k$-gram statistics for various lengths of $k = 1, 2, \ldots, M^{(2)}$ via multi-head attention and then process the the statistics for follow-up optimal scheme.

### D.2.3 INDUCTIVE CTW LAYER

Recall the input and the expected outputs of the inductive CTW layer that

$$\mathbf{h}_i^{(\ell)} = (\mathbf{s}_{i,M^{(1)}+1}; \mathbf{p}_{i,D}; \mathbf{l}_{i,D}^e; \delta_{i,D}; \delta_{i,D-1}; \ldots; \delta_{i,D-\ell+4}; \ell_{i,s_{i,D+3-\ell}}^w; \mathbf{0}; \mathbf{pos}_i), \tag{47}$$

for $\ell = 3, 4, \ldots, 3+D$, where $\delta_{i,l} := \ln(\omega_{i,l}) - \ln(\omega_{i,l-1})$ for $l = d, D-1, \ldots, 1$ are the the weight difference, and we take $M^{(1)} = D$.

**Theorem 9** (Restatement of Theorem 5). *There exists a A-head transformer layer that can perform the induction: Takes $\mathbf{H}^{(\ell)}$ in (47) as input and outputs $\mathbf{H}^{(\ell+1)}$. And the final readout layer taking $\mathbf{H}^{(D+2)}$ as input can output the A-dimensional Bayesian optimal next token prediction vector $P_{\pi_{CTW}}(\cdot|x_{1-D}^n) = \sum_{l=0,\ldots,D}\omega_{n,l}\mathbf{p}_{n,s_{n,l}}.$*

*Proof of Theorem 5.* For any fixed $\ell = 3, 4, \ldots, 2 + D$, we specify the construction for the $\ell$-th transformer layer. It contains $A$ heads and for each $m = 1, 2, \ldots, A$, the $Q, K, V$ matrices are

$$
W_{Q,m}^{(\ell)} = \begin{pmatrix} \mathbf{I}^{(D+1-\ell)A \times (D+1-\ell)A} & \mathbf{0} \\ \mathbf{0} & [\mathbf{e}_m, \mathbf{0}^{A \times 2}] \\ \mathbf{0} & \mathbf{0} \\ \mathbf{0} & \mathbf{I}^{2 \times 2} \end{pmatrix}, \quad W_{K,m}^{(\ell)} = \begin{pmatrix} c\mathbf{I}^{(D+2-\ell)A \times (D+2-\ell)A} & \mathbf{0} \\ \mathbf{0} & \mathbf{0} \\ \mathbf{0} & c\mathbf{I}^{2 \times 2} \end{pmatrix},
$$

$$
W_{V,m}^{(\ell)} = \begin{pmatrix} \mathbf{0}^{(\text{place}_\ell + m) \times (\text{place}_\ell + m)} & \mathbf{0} \\ [\mathbf{0}^{1 \times (\text{place}_\ell - 1)}, 1] & \mathbf{0} \\ \mathbf{0} & \mathbf{0} \end{pmatrix},
$$

where $\mathbf{e}_m$ is the $A$-dimensional one-hot vector at position $m$, and $\text{place}_\ell = (M^{(1)} + D + 2)A + D + \ell - 1$ is index of element $\ell_{i,s_{i,D+3-\ell}}^w$ in $\mathbf{h}_i^{(\ell)}$. The corresponding query, key, and value vectors after the mapping are

$$
W_{Q,m}^{(\ell)} \mathbf{h}_n^{(\ell)} = \begin{pmatrix} \mathbf{s}_{n,D+1-\ell} \\ \mathbf{e}_m \\ \mathbf{0} \\ \mathbf{pos}_n \end{pmatrix}, \quad W_{K,m}^{(\ell)} \mathbf{h}_i^{(\ell)} = c \begin{pmatrix} \mathbf{s}_{i,D+2-\ell} \\ \mathbf{0} \\ \mathbf{pos}_i \end{pmatrix}, \quad W_{V,m}^{(\ell)} \mathbf{h}_i^{(\ell)} = \begin{pmatrix} \mathbf{0}^{(\text{place}_\ell + m) \times 1} \\ \ell_{i,s_{i,D+3-\ell}}^w \\ \mathbf{0} \end{pmatrix}.
$$

At position $n$, the query of $m$-head will select the latest (due to positional embedding) position with suffix $[\mathbf{s}_{n,D+1-\ell}; \mathbf{e}_m]$, and append its $\ell^w$ at the end. It is not hard to see that taking $c \to \infty$ gives

$$
\mathbf{a}_i^{(\ell)} = [\text{MHA}(\mathbf{H}^{(2)}) + \mathbf{H}^{(2)}]_i
$$
$$
= (\mathbf{s}_{i,D+1}; \mathbf{p}_{i,D}; \mathbf{l}_{i,D}^e; \delta_{i,D}; \delta_{i,D-1}; \ldots; \delta_{i,D+4-\ell}; \ell_{i,s_{i,D+3-\ell}}^w; [\ell_{i,qs_{i,D+2-\ell}}^w]_{q \in \mathcal{A}}; \mathbf{0}; \mathbf{pos}_i)
$$

Recall $\ln(\omega_{n,l}) - \ln(\omega_{n,l-1}) = \ln(1-\lambda) - \mathbb{I}_{l=D} \ln(\lambda) + \ell_{n,s_{n,l}}^e - \ell_{n,s_{n,l-1}}^e + \sum_{q \in \mathcal{A}} \ell_{n,qs_{n,l-1}}^w - \ell_{n,s_{n,l}}^w$ by Theorem 2. $\delta_{i,D+3-\ell} = \ln(\omega_{i,D+3-\ell}) - \ln(\omega_{i,D+2-\ell})$ can be computed by $\mathbf{a}_i^{(\ell)}$ and thus $\mathbf{h}_i^{(\ell+1)}$ can be approximated via the FF layer following the $\ell$-th multi-head attention layer.

The final layer approximate an $A$-dimensional vector

$$
P_{\pi_{\text{CTW}}}(\cdot | x_{1-D}^n) = \sum_{l=0,\ldots,D} \omega_{n,l} \cdot \mathbf{p}_{n,s_{n,l}}(\cdot), \tag{48}
$$

by an FF layer taking input

$$
\mathbf{h}_n^{(D+3)} = (\mathbf{s}_{n,M^{(1)}+1}; \mathbf{p}_{n,D}; \mathbf{l}_{n,D}^e; \delta_{n,D}; \ldots; \delta_{n,1}; \mathbf{0}; \mathbf{pos}_i). \tag{49}
$$

The proof is now complete. $\qquad \square$

