# OpenReview forum: "Transformers Learn Variable-order Markov Chains in-Context"
_ICLR.cc/2025/Conference — Submitted to ICLR 2025_

### Official Review · Reviewer_P9pL · 2024-11-04

**Soundness:** 2
**Presentation:** 3
**Contribution:** 2
**Rating:** 5
**Confidence:** 3

**Summary:**

This article examines how transformers learn variable-order MC. It extends previous research on the learning of fixed-order MC.

**Strengths:**

1. The authors found that transformers can learn to compress VOMC in-context, approaching the optimal CTW algorithm for appropriate CTW-priors.
2. The authors found that transformers do not require high network complexity. Even 2-layer transformers can perform well.

**Weaknesses:**

1. Research on fixed-order MC helps us understand how the in-context learning capabilities of LLMs emerge. This is an important finding. Building on this, what key discoveries does this paper offer regarding variable-order MC? The authors' findings, including (1) transformers can learn VOMC in-context capabilities, and (2) transformers are more powerful than CTW, do not seem to bring significant new contributions or explanations.
2. To my understanding, the authors investigated the expressive capabilities of transformers. However, this has already been established. Previous work has found that even transformers with only one encoder layer and three decoder layers are Turing complete. Considering this, what stronger discoveries does the author's research on expressive capabilities bring compared to Turing completeness?
3. Why compare with CTW? Is it a sufficiently powerful framework?
4. What practical insights can be gained from the study of VOMC?

**Questions:**

See weaknesses.

---

> ### Author Response · Authors · 2024-11-21
>
> **Relation to the study of FOMCs** We first emphasize that the VOMC (context tree) model is a widely accepted model in information theory and statistical language modeling, because it is substantially more expressive and aligns better with linguistic phenomena than other simpler models. To see this, let us consider an example sentence ``Language models are useful in a wide \textcolor{blue}{variety} of applications, from natural language understanding and generation to translation, \textcolor{blue}{summarization}, and...''. It is clear the token "variety" depends (almost entirely) on the previous several tokens "useful in a wide" but not other tokens; on the other hand, the token "summarization" probably depends on a much longer suffix, for example perhaps "from natural language understanding and generation to translation,". This dependence on variable length suffixes is a hallmark of natural languages, which fixed-order Markov models cannot capture at all. Past studies such as Edelman 2024 and Akyurek 2024 only studied the fixed-order model, which over-simplifies natural language, implying that they will unavoidably miss important mechanisms to thoroughly understand ICL.
>
> Despite the only change from variable-order to fixed-order, VOMCs are in fact considerably more challenging to analyze, evaluate, to learn, and therefore, it is much more difficult to answer the question of whether transformers can indeed learn it, and if so, how it is done. As explained in the paper, a VOMC model includes an underlying tree that is unknown, and the corresponding unknown statistics. Learning such models is related to "structural learning", in contrast to simple statistical learning. A good analogy is in structural graphical model learning, where there is a huge difference between when the underlying graph structure is known and when the graph is not known but needs to be inferred from the data. Here the situation is very similar. Not knowing the underlying tree makes it exponentially difficult to learn. To put things into perspective: consider a ternary alphabet context tree of maximum order 5, which has over 380 million potential tree configurations; how can transformers identify and leverage relevant structures to perform ICL among such a large number of candidates? How do transformers perform the dual task of structural learning and statistical learning? Can these be done efficiently? In fact, how do we even define "can learn" in this complex setting? These are the difficulties unique in our problem setting. In contrast, for a FOMC, a ternary alphabet of order 5 already specifies the tree, and therefore, the only task is statistical learning of the next token distribution, and an MLE estimator suffices. Therefore, the model we study is much more realistic and significantly more difficult to learn, and the questions of whether transformers can learn and how, are also much more difficult to answer.

---

> > ### Author Response · Authors · 2024-11-21
> >
> > **Regarding the expressibility result:** Our focus is not on expressive completeness but on uncovering the more precise underlying mechanisms through which transformers achieve in-context learning for VOMCs. An unstated goal is to identify the components in transformers that are similar to or different from traditional CTW algorithms, and to do this, the expressive results need to be more functional, rather than general existence-type statements. The Turing complete result mentioned by the reviewer is in fact straightforward to understand in our setting: here the state in the Turing machine can be viewed as the complete history in the context window for each position, and the Turing machine function outputs the next token probability, and update the state by appending the current token. The argument would be that transformers can also do this, because the feedforward network in the transformer can (due to the universal approximation capability of the FF network) mimic this function, which takes the whole history (the Turing machine state) and the current symbol as the input, then outputs the next token probability. The existence statement is true, but it also almost trivializes the transformer model. More precisely, note the following: 1) the existence of such a function does not mean we can construct one explicitly; 2), the feedforward network needs to be extremely wide to approximate such a complex function (directly from history to prediction); 3) our own experiments and past results also suggest that this is not the mechanism transformers use to learn FOMCs or VOMCs; 4) this requires either some rather exotic hidden-space coding mechanism, a very large embedding space, or a large number of attention head, in order to form the state in the hidden vector at each position.
> >
> > Additionally, there were actually three desiderata in our construction even though we did not explicitly state them: 1) Do not use exotic hidden-space coding, 2) Do not use too many attention heads (e.g., $|A|^D$), and 3) Do not use a large number of layers (e.g., somehow related to the context window size). These are quite realistic requirements we believe, however, the construction becomes rather constrained, and that is why our proposed construction is quite sophisticated and highly non-trivial.

---

> > > ### Author Response · Authors · 2024-11-21
> > >
> > > **Rationale to compare to CTW:**  Consider an analogy of the question whether "transformers can learn linear regression", where the optimal linear regression solution must be compared with to determine the answer. In order to claim "Transformers can learn VOMCs", we have to compare it to the optimal performance as well. What is the optimal performance here? It turns out CTW is Bayesian optimal in this setting and thus the right benchmark to compare the transformers' performance with. Transformers can surpass CTW algorithm's performance under a different prior. By analyzing transformers from the CTW perspective, we can then meaningfully evaluate whether "transformers can learn VOMCs", and the CTW is also then the optimal algorithm for the transformers to target and approximate.
> > >
> > >
> > > **Regarding the practical insights:** First we partially answer the question in the item "Relation to the study of FOMCs". In terms of practical importance, we did have some insights that may benefit more complex LLMs design, or conversely, benefit the design of advanced compression algorithms. We note that these points would require further research that is beyond the scope of this work. (1) The lower-layer architecture of building suffixes can be viewed as an important component of the lower-layer functionality. Therefore, a transformer design of having some of these fixed components together with other trainable heads may lead to better training or improved performance. (2) The hybrid model suggests the counting mechanism in a certain manner plays an important role, but transformers are not very efficient at counting. We may then attempt a design that inserts such counting information directly in some lower transformer layers, instead of hoping the trained attention heads find them through training. (3) The FF layers seem to be important for upper-layer functions. Therefore, a large model that uses narrower FF layers at the lower transformer layers, and wider and deeper FF layers at higher transformer layers may be beneficial, even resulting in the saving of model parameters. (4) In the other direction, the insight may help the design of new compression algorithms. The CTW algorithm only uses suffixes, whereas we found that the suffix representation in transformers is much more flexible; See Fig 6, middle panel. This flexibility appears to be a more efficient way that could allow CTW to proceed beyond the simple suffix space. We can add some discussion on these points if the paper is eventually accepted.

---

### Official Review · Reviewer_HVxK · 2024-11-04

**Soundness:** 3
**Presentation:** 2
**Contribution:** 3
**Rating:** 3
**Confidence:** 3

**Summary:**

The paper explores the in-context learning (ICL) capabilities of large language models (LLMs) by introducing a novel perspective of viewing language modeling as data compression. The study finds that transformers can effectively learn to compress VOMC in-context, outperforming traditional compression algorithms under certain conditions. Key phenomena observed include the resilience of transformers to layer variations and their ability to surpass CTW algorithms, particularly when non-CTW priors are used. The paper also provides theoretical frameworks to explain these observations.

**Strengths:**

- The paper presents an innovative perspective on understanding ICL by framing it as a context information compression task, which broadens the theoretical understanding of transformer capabilities.
- It offers empirical evidence and theoretical constructions that explain the observed phenomena (like Fig 6&7), providing insights into how transformers can mimic and even surpass traditional compression methods.

**Weaknesses:**

- The work attempts to interpret ICL, but for readers familiar with ICL/LLM, the introduction of transformer attention (Sec 2.1) is somewhat redundant. Conversely, there is insufficient background information on CTW and related work, which are crucial for understanding the paper's context.
- The paper is challenging to follow without consulting the appendix, like Appendix A/B, which is essential for understanding the introduction and foundational concepts.
- There is a lack of real-world ICL example analyses, especially considering that the data used in the paper is different from the real data used in LLM ICL. Incorporating such examples, especially if the real ICL attention pattern mirrors those in Figures 6 and 7, would provide more robust validation.

**Questions:**

- How is the compression rate computed, and why can this rate exceed 1 for the PPM algorithm, as shown in Figure 4?
- How much of the training data in Section 3.1 is constructed, is the data sufficient enough for training, and what are the details of the training process, like the learning rate?
- What does the term "segment index" refer to in Figures 4 and 5?
- Typo: Line 356: "Fig 7" should be corrected to "Fig 6"

---

> ### Author Response · Authors · 2024-11-21
> **Response to Reviewer HVxK weaknesses**
>
> **Redundancy in Section 2.1 (Transformer Attention):**
> We understand that the introduction of Transformer attention (Section 2.1) may appear redundant for readers familiar with in-context learning (ICL) and large language models (LLMs). However, our goal in including this section is to ensure accessibility for readers from diverse backgrounds, especially those more familiar with statistical modeling (e.g., context trees) but less familiar with Transformer mechanisms.
>
> **Insufficient Background on CTW and Related Work:**
> We appreciate the reviewer’s feedback regarding the background of the Context Tree Weighting (CTW) algorithm. We have to strike a balance between providing necessary technical details and maintaining focus on the core contributions of the paper. We included the essential information on the CTW algorithm in the main text, such that readers familiar with this topic or equipped with the included material should be able to understand its role in the study. In fact, even if we have provided a very thorough introduction to CTW, a complete understanding of the CTW algorithm is unlikely, given the complex computation structure of the CTW algorithm. Instead, we made the conscientious choice of providing the most important descriptions that are related to the development of the theoretical results needed for our work. These descriptions, though brief, should be sufficient to develop the theory and verify the correctness. If the paper is accepted, We will add more details to help readers who are interested in the technical details, but again, those details are not needed for the theory development in our work.
>
> **Difficulty Following the Paper Without Consulting the Appendix:**
> We appreciate the feedback on the paper’s structure. Given the page limit, we have to strike a balance somewhere. After several rounds of revisions before submission, we feel the information in the introduction and preliminary provides enough knowledge to understand the framework, the development, and the results. Additional information will have to be kept in the appendix. We would really appreciate it if the review could provide more details on what specific information is of vital importance, and should be promoted to the main text.
>
> **Lack of Real-World ICL Example Analyses:**
> The primary objective of this study is to develop a clear understanding of ICL for more complex settings. In order to answer the question clearly, we must restrict ourselves to a "clean environment". We therefore focus on Variable-Order Markov Chains (VOMCs) as a simplified yet representative probabilistic model for language structure. The experimental setup is intentionally designed to align with the theoretical framework, validating our analysis under controlled conditions. In a sense, we are continuing along the line of the "induction head" research (Olsson et al. ) started by a group of researchers in Anthropic on understanding how transformers really learn. Many questions were not clearly answered in that work, exactly because the study was empirical on real languages. In contrast, our work is in a "clean lab environment", which leads us to more precise results and understanding. We believe the Anthropic team was able to use their understanding in their model design, and our findings can provide more insights. This is reminiscent of biology studies: both in-vivo studies and in-vitro studies can be used for the same problem, but in-vitro can provide clean answers in a well-controlled lab environment, and applying them in-vivo may require additional adaptation. However, we would never question the value of in-vitro work in biology.

---

> > ### Author Response · Authors · 2024-11-21
> > **Response to Reviewer HVxK questions**
> >
> > **Compression Rate in Figure 4 and PPM Algorithm:**
> > The compression rate in Figure 4 is the same as cross-entropy loss. The range of cross-entropy is all non-negative values and can be greater than 1. We will clarify this in the caption of Figure 4 and the corresponding section.
> >
> > **Details of Training Data in Section 3.1:**
> > Please refer to Appendix C. The number of data is $K = 20000$ CTs. The amount of data is sufficient for training as the ICL performance is comparable to the optimal performance. Transformers are trained using AdamW optimizer with default parameters.
> >
> > **Segment Index in Figures 4 and 5:**
> > We divide the source sequence into multiple segments, where each segment contains $16$ or $48$ tokens, depending on the context window length. These details can be found in the appendix. We can move some details forward if the reviewer feels it is very important.
> >
> > **Typo in Line 356:**
> > Thank you for pointing this out. We will correct "Fig 7" to "Fig 6" in the revised manuscript.

---

> > > ### Comment · Reviewer_HVxK · 2024-11-24
> > >
> > > Thank you for your response. While this work may interest researchers familiar with context trees, I believe that most of the audience is more knowledgeable about LLMs and transformers, as LLMs are currently a hot topic and ICL, as a crucial skill of LLMs, its interpretability is an intriguing subject. You can consider incorporating additional perspectives on data compression as motivation in the introduction. Furthermore, the lack of analysis on real ICL examples weakens your conclusion that ICL is not sensitive to the number of layers because this conclusion is only valid under this toy setting. I will keep my score.

---

### Official Review · Reviewer_oXGR · 2024-11-10

**Soundness:** 2
**Presentation:** 2
**Contribution:** 2
**Rating:** 5
**Confidence:** 3

**Summary:**

This paper explores the in-context learning (ICL) capabilities of transformers for variable-order Markov chains (VOMC), viewing language modeling as a data compression problem. The authors benchmark transformer models against compression algorithms like context-tree weighting (CTW) and prediction by partial matching (PPM). Key findings include: (1) transformers effectively learn VOMC in context, outperforming PPM; (2) ICL performance is relatively insensitive to layer count, with even a two-layer transformer performing well; and (3) transformers can surpass CTW on non-CTW priors. The authors propose two transformer architectures to mimic CTW and explain observed behaviors, with a focus on the role of counting mechanisms in ICL.

**Strengths:**

1. The study’s attempt to connect data compression techniques with transformer-based models is innovative and provides new insights into LLM behavior in sequence modeling.
2. The derivation of two specialized transformer structures based on CTW and PPM is well-motivated.

**Weaknesses:**

1. The experimental setup in the paper is limited, lacking comprehensive benchmarks for language modeling. More diverse and practical tasks, such as question answering and natural language understanding, should be used to confirm the robustness of the proposed transformer models.
2. The experiments were not validated on larger-scale pre-trained language models, such as models of the scale of GPT or LLaMA, which limits the study's practical relevance. Additionally, there is a lack of discussion on how these findings could be applied to mainstream LLM tasks.
3. The paper's readability could be improved.

**Questions:**

please refer to weakness

---

> ### Author Response · Authors · 2024-11-21
> **Response to Reviewer oXGRv**
>
> **Regarding benchmark:**
> The primary objective of this study is to develop a clear understanding of ICL for more complex settings. In order to answer the question clearly, we must restrict ourselves to a "clean environment". We therefore focus on Variable-Order Markov Chains (VOMCs) as a simplified yet representative probabilistic model for language structure. The experimental setup is intentionally designed to align with the theoretical framework, validating our analysis under controlled conditions. In a sense, we are continuing along the line of the "induction head" research (Olsson et al. ) started by a group of researchers in Anthropic on understanding how transformers really learn. Many questions were not clearly answered in that work, exactly because the study was empirical on real languages. In contrast, our work is in a "clean lab environment", which leads us to more precise results and understanding. We believe the Anthropic team was able to use their understanding in their model design, and our findings can provide more insights. This is reminiscent of biology studies: both in-vivo studies and in-vitro studies can be used for the same problem, but in-vitro can provide clean answers in a well-controlled lab environment, and applying them in-vivo may require additional adaptation. However, we would never question the value of in-vitro work in biology.
>
> **Regarding testing on pre-trained models:**
> Regarding the validations on mainstream LLMs: While we recognize the importance of diverse and practical benchmarks, our goal in this work is not to benchmark state-of-the-art language models but to provide a conceptual and theoretically grounded understanding of ICL mechanisms. In fact, using pre-trained LLMs can not provide an answer to the key question of whether transformers can learn VOMCs. This is because the problem is best studied from a Bayesian perspective, however, there is no way to control the data priors in the training of those LLMs, and testing on them can only lead to misleading answers.
>
> **Regarding the readability:**
> Regarding the readability: we would be happy to modify based on specific readability issues raised.

---

### Official Review · Reviewer_m9fp · 2024-11-11

**Soundness:** 4
**Presentation:** 3
**Contribution:** 1
**Rating:** 3
**Confidence:** 4

**Summary:**

This paper investigates in-context learning ability of auto-regresive transformers on data generated by variable order markov chains (VOMC). Previous work Edelman 2024 investigated fixed order markov chains (~ngram languages), Akyurek 2024 investigated uniform hidden markov models (probabilistic regular languages), and this is in the same line but learning a different class of probabilistic languages (i.e. probabilistic models). The paper experimentally shows the solutions of transformers are better than the Context Tree Weighting algorithm which is the optimal learning algorithm for a subset of VOMC problems. The paper then goes into proofs showing that the Transformer is theoretically capable of running CTWs.

**Strengths:**

- The paper is math heavy but it has high readability.
- The paper finds ICL on VOMC less dependent on number of layers, which is interesting.
- Experiments guide theory to construct a manually crafted Transformer that gets better performance in this VOMC dataset

**Weaknesses:**

- The paper does not explain why studying VOMC has any practical importance — what do we learn about Transformers? What is the additional information gained after Edelman 2024, Akyurek 2024 as they studied in-context learning of very similar probabilistic models.

- The paper has a very similar structure to Akyurek 2022, Akyurek 2024, Edelman 2024 with a slight change to the class of probabilistic models used. This hurts the novelty aspect of this paper.

- There is a long theory section for two different construction, I would keep only 4.3. I would move them 4.2 to the appendix and display the main theorems only.

**Questions:**

I think this paper is sound! But it might be of interest to a specific, purely theoretical ML audience, and I do not think it contributes significantly to the existing work. Therefore, unfortunately I lean towards reject, and I suggest authors to consider a motivation and show that their findings can have some practical importance, otherwise the writing, theory and analysis seems complete to me.

---

> ### Author Response · Authors · 2024-11-21
> **Practical Importance and Novelty, and Contribution Beyond Previous Works**
>
> We thank the reviewer for recognizing the strength of our work.
>
> **Practical Importance and Novelty, and Contribution Beyond Previous Works:** We first emphasize that the VOMC (context tree) model is a widely accepted model in information theory and statistical language modeling, because it is substantially more expressive and aligns better with linguistic phenomena than other simpler models. To see this, let us consider an example sentence ``Language models are useful in a wide variety of applications, from natural language understanding and generation to translation, summarization, and...''. It is clear the token "variety" depends (almost entirely) on the previous several tokens "useful in a wide" but not other tokens; on the other hand, the token "summarization" probably depends on a much longer suffix, for example perhaps "from natural language understanding and generation to translation,". This dependence on variable length suffixes is a hallmark of natural languages, which fixed-order Markov models cannot capture at all. Past studies such as Edelman 2024 and Akyurek 2024 only studied the fixed-order model, which over-simplifies natural language, implying that they will unavoidably miss important mechanisms to thoroughly understand ICL.
>
> Despite the only change from variable-order to fixed-order, VOMCs are in fact considerably more challenging to analyze, evaluate, to learn, and therefore, it is much more difficult to answer the question of whether transformers can indeed learn it, and if so, how it is done. As explained in the paper, a VOMC model includes an underlying tree that is unknown, and the corresponding unknown statistics. Learning such models is related to "structural learning", in contrast to simple statistical learning. A good analogy is in structural graphical model learning, where there is a huge difference between when the underlying graph structure is known and when the graph is not known but needs to be inferred from the data. Here the situation is very similar. Not knowing the underlying tree makes it exponentially difficult to learn. To put things into perspective: consider a ternary alphabet context tree of maximum order 5, which has over 380 million potential tree configurations; how can transformers identify and leverage relevant structures to perform ICL among such a large number of candidates? How do transformers perform the dual task of structural learning and statistical learning? Can these be done efficiently? In fact, how do we even define "can learn" in this complex setting? These are the difficulties unique in our problem setting. In contrast, for an FOMC, a ternary alphabet of order 5 already specifies the tree structure, and therefore, the only task is statistical learning of the next token distribution, and an MLE estimator suffices. Therefore, the model we study is much more realistic and significantly more difficult to learn, and the questions of whether transformers can learn and how, are also much more difficult to answer.
>
> A key insight gained from our study is the analysis of how Transformers learn in-context to count statistics across multiple granularities -- ranging from 1-gram to (D+1)-gram -- when exposed to sequences generated by VOMCs. This reveals the mechanisms by which Transformers adapt to language structures beyond fixed-order dependencies, which is a critical step toward understanding ICL in real-world scenarios. While the structure of our theoretical analysis shares some similarities with prior works, such similarities are expected given the shared goal of analyzing ICL in Transformers. The difficulty hidden in our construction. However, changing the class of probabilistic models—from FOMCs to VOMCs—introduces novel technical challenges that require us to develop new theoretical tools and insights. For example, due to the unknown structure of the context trees, an optimal blending from 1-gram to (D+1)-gram statistics is required to make optimal predictions, as illustrated in Theorem 2. Overall, our result marks a major step toward understanding how to learn ICL in more realistic and complex settings beyond previous results.

---

> > ### Author Response · Authors · 2024-11-21
> > **Structure of the Theory Section and Target Audience**
> >
> > **Structure of the Theory Section:**
> > We understand the reviewer’s concern about the length of the theoretical section. However, we believe that Sections 4.2 and 4.3 serve distinct purposes: Section 4.2 provides the concept of optimal blending, which is a key to optimal ICL of VOMCs and foundational construction to achieve that, while Section 4.3 focuses on providing more experimental evidence to the theoretical findings.
> >
> > **Target Audience and Broader Impact**
> > While our work is indeed theoretical, its implications extend beyond a purely theoretical audience. In a sense, we are continuing along the line of the "induction head" research (Olsson et al. 2022) started by a group of researchers in Anthropic on understanding how transformers really learn. Many questions were not clearly answered in that work, exactly because the study was empirical on real languages. In contrast, our work is in a "clean lab environment", which leads us to more precise results and understanding. We believe the Anthropic team was able to use their understanding in their model design, and our findings can provide more insights. This is reminiscent of biology studies: both in-vivo studies and in-vitro studies can be used for the same problem, but in-vitro can provide clean answers in a well-controlled lab environment, and applying them in-vivo may require additional adaptation. However, we would never question the value of in-vitro work in biology. By analyzing VOMCs, we offer insights into the Transformer’s potential to generalize across a broader range of probabilistic structures. This understanding is not only of theoretical interest but also informs practical applications, such as language modeling, hierarchical sequence prediction, and adaptive learning in diverse domains where the underlying data structure exhibits variable-order dependencies. Understanding in-context learning (ICL) for natural language from a theoretical perspective is still in the nascent stage. We believe our work is an important step towards the overarching goal.
> >
> >
> > In terms of practical importance, we did have some insights that may benefit LLMs design, or conversely, benefit the design of advanced compression algorithms. We note that these insights would require further research that is beyond the scope of this work. (1) The lower-layer architecture of building suffixes can be viewed as an important component of the lower-layer functionality. Therefore, a transformer design of having some of these fixed components together with other trainable heads may lead to better training or improved performance. (2) The hybrid model suggests the counting mechanism in a certain manner plays an important role, but transformers are not very efficient at counting. We may then attempt a design that inserts such counting information directly in some lower transformer layers, instead of hoping the trained attention heads find them through training. (3) The FF layers seem to be important for upper-layer functions. Therefore, a large model that uses narrower FF layers at the lower transformer layers, and wider and deeper FF layers at higher transformer layers may be beneficial, even resulting in saving in model parameters. (4) In the other direction, the insight may help the design of new compression algorithms. The CTW algorithm only uses suffixes, whereas we found that the suffix representation in transformers is much more flexible; See Fig 6, middle panel. This flexibility appears to be a more efficient way that could allow CTW to proceed beyond the simple suffix space. We can add some discussion on these points if the paper is eventually accepted.

---

### Meta-Review · Area_Chair_m7Jp · 2024-12-22

**Metareview:**

Based on the reviewers' feedback, I recommend rejecting this paper at this time. While the paper presents an interesting analysis of transformers' in-context learning capabilities for variable-order Markov chains through a data compression lens, several critical issues emerge. First, the paper's novelty is limited, as R1 points out that it largely follows the structure and approach of previous works Akyurek 2022, Akyurek 2024, Edelman 2024 with a slight change to the class of probabilistic models used. Second, the practical significance and insights gained from studying VOMC remain unclear, especially given that transformers' expressive capabilities are already well-established. Third, the experimental validation is limited, lacking comprehensive benchmarks and real-world ICL examples that would demonstrate practical relevance. While the mathematical treatment is sound and the paper is generally well-written, these fundamental concerns about novelty, practical relevance, and limited experimental validation make it difficult to justify acceptance.

**Additional Comments On Reviewer Discussion:**

I have read the messages in the discussion period and my opinion has been summarized as in the metareview above. I considered these points in my recommendation.

---

### Decision · Program_Chairs · 2025-01-22

Reject